# Where to Begin: Efficient Pretraining via Sub-network Selection and Distillation

## Abstract

Small Language models (SLMs) offer an efficient and accessible alternative to Large Language Models (LLMs), delivering strong performance while using far fewer resources. We introduce a simple and effective framework for pretraining SLMs that brings together three complementary ideas. First, we identify structurally sparse **sub-network initializations** that consistently outperform randomly initialized models of similar size under the same compute budget. Second, we use **evolutionary search** to automatically discover high-quality sub-network initializations, providing better starting points for pretraining. Third, we apply **knowledge distillation** from larger teacher models to speed up training and improve generalization. Together, these components make SLM pretraining substantially more efficient: our best model, discovered using evolutionary search and initialized with LLM weights, matches the validation perplexity of a comparable Pythia SLM while requiring $5.16\times$ and $1.26\times$ fewer floating point operations for token budgets of 10B and 100B, respectively. We release all code publicly, offering a practical and reproducible path toward cost-efficient small language model development at scale.

## 1 Introduction

Large Language Models (LLMs) have recently delivered state-of-the-art performance across a wide range of tasks. Their success is largely driven by scale: modern LLMs routinely exceed tens and hundreds of billions of parameters, unlocking remarkable generalization and emergent abilities. However, this scale comes at a cost. Training and deploying such massive models requires substantial computational resources, and inference often exceeds practical memory or latency budgets.

These challenges have motivated increasing interest in **Small Language Models (SLMs)** (Allal et al., 2025; Yang et al., 2025), which aim to preserve strong performance while remaining deployable in resource-constrained settings such as mobile or edge devices. Although pretraining SLMs is substantially cheaper than training LLMs, the costs are still formidable and often beyond the reach of most smaller research groups. For example, Allal et al. (2025) estimate that training SmolLM2 with 1.7B parameters required on the order of $10^{23}$ FLOPs—roughly \$250,000 of GPU compute.

A common strategy to reduce pretraining cost is to leverage open-weight LLMs as teachers. For instance, Team et al. (2025) used knowledge distillation to train the Gemma 3 family. This idea can be pushed further by warm-starting students from non-random initializations derived from their teachers. Muralidharan et al. (2024) demonstrated this by pruning a teacher model and refining it through distillation, while the smaller variants of Llama 3.2 (Meta AI, 2024) were similarly obtained using a combination of pruning and distillation.

Unfortunately, most existing efforts in this space are closed-source, making them difficult to reproduce and extend. While the evidence so far suggests that teacher models can greatly improve the efficiency of SLM pretraining, the underlying mechanisms remain poorly understood. In this work, we present the first systematic open-source study of *warm-starting student models from larger teachers for pretraining*. Our contributions are:

- **Sub-network initialization.** We propose a new warm-starting strategy that extracts high-quality sub-networks from pretrained teachers. The smaller variants (around 410M parameters) require $1.71\times$ fewer FLOP of pretraining than a comparable Pythia-410M model to achieve the same validation perplexity. Larger variants achieve higher speed-ups, with

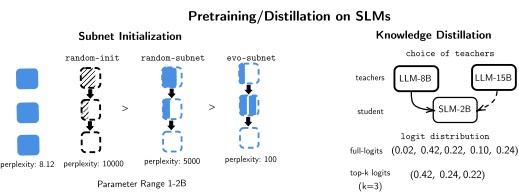

Figure 1: Left: Initialization schemes — random weights, sub-network from a pretrained teacher, and our evolutionary search–based sub-network. Right: The same teacher is used for knowledge distillation to train the student.

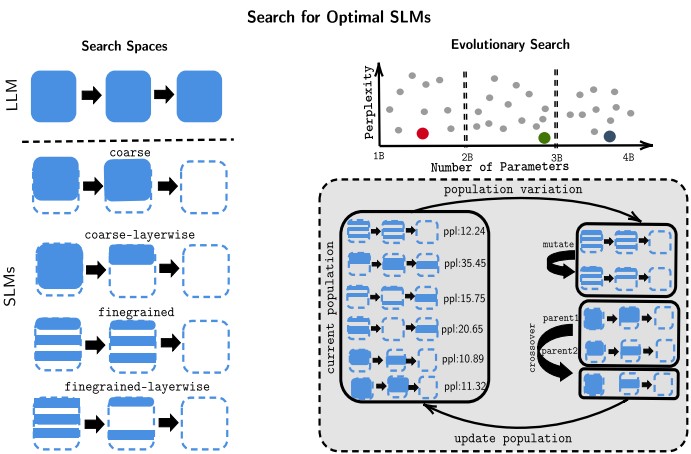

Figure 2: Overview of our search spaces and search strategy

models comparable to Pythia-1B requiring 1.75× fewer FLOP using pretraining, and models comparable to Pythia-2.8B requiring 5.16× fewer FLOP during pretraining (see Appendix B.1 for details). We also analyze how different search spaces and extraction strategies affect downstream performance.

- **Comprehensive analysis.** We provide the first systematic comparison of sub-network initialization under knowledge distillation versus standard cross-entropy training, showing the benefits of knowledge distillation over standard pretraining. Our study spans multiple student scales and investigates how teacher size influences effectiveness of distillation.

- **Reproducible framework.** We release an open-source library[1] for extracting sub-networks from existing LLM checkpoints. Together with our empirical findings, this establishes practical guidelines for compute-optimal SLM pretraining across different scales.

Section 2 presents our methodology for extracting sub-networks from a pretrained teacher network, and Section 3 introduces our open-source library for sub-network extraction. We provide an empirical analysis and compare to baseline approaches in Section 4. In Appendix A, we discuss prior work relevant to our approach.

## 2 METHODOLOGY

We study the problem of pretraining a Small Language Model (SLM) with the help of a larger open-weight teacher. Our approach follows a two-step strategy: (i) extract a sub-network from the pretrained teacher, and (ii) use this sub-network as initialization for SLM pretraining with knowledge distillation. In this section, we describe the key components of this pipeline. We first introduce the search space granularities considered (Section 2.1), then present our constrained evolutionary search procedure (Section 2.2), and finally delineate the pretraining and distillation process (Section 2.3).

### 2.1 SEARCH SPACES

We consider a dense transformer model $T$, with $L$ layers and embedding dimension $E$. Each layer $i \in 1, \ldots, L$ consists of a causal self-attention block with $H$ attention heads of dimension

---

[1] https://anonymous.4open.science/r/whittle-iclr-71CD/

$H_s$, followed by an MLP block with intermediate dimension $D$. For simplicity, we restrict our discussion to the multi-head attention setting, though the approach extends naturally to multi-query and group-query attention.

We parameterize a sub-network $S$ of the teacher model $T$ by specifying the number of layers $l \in 1, \ldots, L$, the embedding dimension $e \in 1, \ldots, E$, the number of attention heads $h \in 1, \ldots, H$, the head dimension $h_s \in 1, \ldots, H_s$, and the MLP intermediate size $d \in 1, \ldots, D$. We define four search spaces that differ in how weights are selected: *coarse* versus *fine-grained*, and *uniform* versus *layer-wise*.

**Coarse.** To construct the sub-network, we always select the *first* $n$ entries from the corresponding components of the teacher $T$. For example, selecting $h$ attention heads corresponds to taking the first $h$ heads out of the $H$ heads in $T$. Likewise, choosing a smaller embedding dimension $e$ corresponds to taking the first $e$ elements of the embedding vector.

**Fine-grained.** We select a subset of size $n$ by sampling *indices* from the teacher's components. For instance, selecting $h$ attention heads corresponds to sampling $h$ distinct heads from the $H$ available in $T$ (without replacement).

Next, we distinguish between two types of layer configurations:

**Uniform.** The same configuration $(h, h_s, d)$ for heads, head size and intermediate MLP size, is applied across all layers. That is, every layer uses the same number of heads, query groups, head dimension, and MLP intermediate size.

**Layer-wise.** Each layer is allowed to have its own configuration, relaxing the uniformity constraint.

Combining the two sampling strategies (*coarse* vs. *fine-grained*) with the two configuration schemes (*uniform* vs. *layer-wise*) yields four distinct search spaces:

**Coarse Uniform.** This is the simplest search space, in which the same configuration is applied to all layers, always selecting the first entries. For multi-head attention layers, the total number of possible configurations is $N = L \cdot E \cdot H \cdot H_s \cdot D$. In the case of group-query attention, $N$ additionally accounts for the number of valid combinations of heads $h$ and query groups $q$.

**Coarse Layer-wise.** The *coarse layer-wise* search space applies coarse sampling independently to each layer in the sub-network $S$, allowing each layer to have its own configuration. The total number of configurations is $N = E \cdot (H \cdot H_s \cdot D)^L$, which grows exponentially with the number of layers $L$. Compared to the *coarse uniform* space, which is linear in $L$, the *coarse layer-wise* space is significantly larger, as each layer can independently select its $(h, h_s, d)$ configuration.

**Fine-grained Uniform .** The *fine-grained uniform* search space applies fine-grained sampling uniformly across all layers. In this setting, the sub-network may be formed from an arbitrary subset of elements within each layer, rather than being restricted to the first $l$ layers. The total number of configurations in this search space is $N = 2^{E \cdot H \cdot H_s \cdot D \cdot L}$.

**Fine-grained Layer-wise.** The *layer-wise fine-grained* search space applies fine-grained sampling independently to each layer, yielding the most granular search space considered. Each layer can independently select its number of heads, query groups, head dimension, and MLP intermediate size, and the sub-network may include an arbitrary subset of layers. The total number of configurations is

$$N = 2^E \cdot (2^H \cdot 2^{H_s} \cdot 2^D)^L \cdot 2^L,$$

which grows exponentially with both the width $(E, H, H_s, D)$ and the depth $L$, making it the largest and most expressive search space among the variants considered.

## 2.2 Evolutionary Search

Before outlining our search procedure, we first formalize our experimental setup.

Let $\mathcal{M}$ denote a large language model (LLM) parameterized by $\boldsymbol{\theta}$, with total parameter count $|\boldsymbol{\theta}| = S$ (in billions). We assume $S > 1$ and typically consider models where $S > 2$. The user specifies a *parameter bin*

$$\mathcal{B} = [S_{\min}, S_{\max}],$$

which defines the range of acceptable model sizes (e.g., $S_{\min} = 1B$, $S_{\max} = 2B$).

We partition the overall parameter space

$$\mathcal{S} = \{\boldsymbol{\theta} : |\boldsymbol{\theta}| \in [S_{\min}^{(i)}, S_{\max}^{(i)}]\}_{i=1}^{K}$$

into $K$ disjoint bins $\{\mathcal{B}_1, \ldots, \mathcal{B}_K\}$, each corresponding to a contiguous range of parameter counts. This stratification ensures balanced coverage across different model sizes. Without such binning, uniform random sampling tends to under-represent very small and very large models.

We now delineate our constrained evolutionary search procedure.

**Evolutionary search with constraint enforcement.** Within each bin $\mathcal{B}_i$, we perform an evolutionary search over candidate sub-network architectures $\mathcal{A}(\boldsymbol{\theta})$. At each iteration, candidate architectures are sampled and evaluated according to a fitness function $f(\mathcal{A})$. To enforce the bin constraint, we apply *rejection sampling*:

$$\mathcal{A} \leftarrow \begin{cases} \mathcal{A}, & \text{if } |\boldsymbol{\theta}_{\mathcal{A}}| \in \mathcal{B}_i, \\ \text{reject}, & \text{otherwise} \end{cases}$$

Only candidates satisfying $|\boldsymbol{\theta}_{\mathcal{A}}| \in \mathcal{B}_i$ are retained for further evolution.

After convergence, we return the set of small language models (SLMs)

$$\mathcal{A}_i^* = \arg\max_{|\boldsymbol{\theta}_{\mathcal{A}}| \in \mathcal{B}_i} f(\mathcal{A}),$$

that achieve the most favorable initialization for subsequent pretraining or distillation. These selected SLMs represent the optimal sub-network architectures within the specified parameter range.

The overall procedure is summarized in Algorithm 1 (Appendix C.2). Within each parameter-size bin, we initialize a population of sub-network candidates drawn from the constrained search space. At each epoch, candidates are evaluated by perplexity and the top-$k$ elites are retained. Genetic operators—**mutation** and **crossover**—then generate offspring subject to bin constraints, while additional random samples encourage exploration. The next population is formed by selecting the $N$ best candidates among elites, offspring, and random samples. After $T$ epochs, the best-performing sub-network in each bin is returned, with mutation and crossover formally defined below.

**Mutation.** Given a candidate architecture $\mathcal{A}$, we define a mutation operator $\mu(\mathcal{A})$ that perturbs one architectural dimension at a time. Specifically, we uniformly sample a dimension

$$x \in \{l, e, h, g, d, h_s\},$$

where $l$ denotes the number of layers, $e$ the embedding dimension, $h$ the number of attention heads, $g$ the number of query groups, $d$ the intermediate (feedforward) dimension, and $h_s$ the per-head size.

**Mutation in layer-wise search space.** In *layer-wise* search spaces, architectural attributes $(h, g, d, h_s)$ are defined independently for each layer. A mutation of the layer count $l \rightarrow l'$ is handled as follows:

$$\mathcal{A}' = \begin{cases} \mathcal{A} \cup \text{newly sampled}(l' - l)\text{layers}, & \text{if } l' > l, \\ \mathcal{A} \setminus \text{last } (l - l') \text{ layers}, & \text{if } l' < l \end{cases}$$

If $x \in \{h, g, d, h_s\}$, we first sample a layer index $i \sim \text{Uniform}\{1, \ldots, l\}$, then resample the chosen dimension for that layer:

$$x_i' \sim \text{Uniform}\{\text{choices}(x)\}$$

Where $\text{choices}(x)$ denotes the valid choices for an architectural attribute $x$ defined by a search space. All other architectural parameters remain fixed.

In the *fine-grained* setting, mutations operate at the neuron level. For instance, mutating the embedding dimension $e \rightarrow e'$ corresponds to

$$\text{if } e' > e: \text{ sample } (e' - e) \text{ new neurons;} \quad \text{if } e' < e: \text{ prune the last } (e - e') \text{ neurons.}$$

This formulation enables smooth exploration of architectures across both coarse (layer-wise) and fine-grained structural variations, while maintaining consistency with the model size constraint.

**Crossover.** To produce a child architecture from two parents, $P_1$ and $P_2$, we apply a crossover operator $\chi(P_1, P_2)$. We first require both parents to share the same number of layers:

$$l_{P_1} = l_{P_2} = l$$

Let the architectural dimensions of each parent be

$$P_1 = (e_1, h_1, g_1, h_{s,1}, d_1), \quad P_2 = (e_2, h_2, g_2, h_{s,2}, d_2),$$

where $e$, $h$, $g$, $h_s$, and $d$ denote the embedding dimension, number of attention heads, number of query groups, head size, and intermediate dimension, respectively.

A child architecture $c$ is then generated by independently inheriting each dimension from one of the two parents:

$$x_c = \begin{cases} x_1, & \text{with probability } 0.5, \\ x_2, & \text{with probability } 0.5, \end{cases} \quad \text{for each } x \in \{e, h, g, h_s, d\}$$

For example, a valid crossover outcome might be

$$c = (e_2, h_1, g_2, h_{s,2}, d_1)$$

This independent dimension-wise crossover enables fine-grained recombination of architectural traits while preserving structural compatibility (e.g., layer count consistency) between parents.

## 2.3 SLM Pretraining and Distillation

**Sub-network Extraction.** Our constrained evolutionary search Algorithm 1 (Appendix C.2), returns a sub-network configuration $s_b$, for every parameter bin $b$. Given this sub-network configuration, we extract the smaller language model corresponding to this configuration from the larger base model we perform search on. We then convert this extracted sub-network into a dense language model with the corresponding architecture. This is then the SLM that we use in our pretaining pipeline, optimizing the standard token-level cross entropy, language modeling loss.

**Knowledge Distillation.** Model distillation (Hinton et al., 2015), or knowledge distillation, compresses a large *teacher* model into a smaller *student* network that achieves similar performance with fewer resources. Instead of training solely on hard labels, the student leverages *soft labels* from the teacher, obtained via temperature-scaled softmax:

$$p_i^{(T)}(z) = \frac{\exp\left(\frac{z_i}{T}\right)}{\sum_j \exp\left(\frac{z_j}{T}\right)}, \tag{1}$$

where $z = [z_1, z_2, \ldots, z_n]$ are logits and $T > 0$ is the temperature. The student parameters $\hat{\mathbf{w}}_\theta$ are optimized with a loss combining hard-label cross-entropy and distillation:

$$\mathcal{L} = \alpha \, \mathcal{L}_{\text{CE}}(\mathbf{y}, \mathbf{s}) + \beta \, \mathcal{L}_{\text{D}}(p_t, p_s), \tag{2}$$

where $\alpha, \beta \in [0, 1]$ ,and $p_t$ and $p_s$ are the teacher and student logit distributions, respectively. In our setting, $\mathcal{L}_{\text{D}}$ is the forward KL divergence,

$$\mathcal{L}_{\text{D}} = \sum_i p_{t_i}{}^{(T)} \log \frac{p_{t_i}{}^{(T)}}{p_{s_i}{}^{(T)}}, \tag{3}$$

encouraging the student distribution $p_s^{(T)}$ to match the teacher's softened distribution $p_t^{(T)}$. In our final knowledge-distillation setup, we use equation 5, with $\mathcal{L}_{\mathcal{D}}$, corresponding to the forward-kl divergence depicted in equation 3.

**Top-$k$ Logit Distillation.** We define a variant of knowledge distillation that truncates the teacher distribution to its $k$ most salient outputs. Let $z_t, z_s \in \mathbb{R}^C$ denote the teacher and student logits, respectively, and $T$ the temperature parameter. The teacher distribution is given by $p_t = \text{softmax}(z_t/T)$. We denote by $\mathcal{K} \subset \{1, \ldots, C\}$ either the indices of the top-$k$ logits of $z_s$, or a subset sampled from $p_t$. The distillation loss is then defined as:

$$\mathcal{L}_{\text{top-}k} = \sum_{i \in \mathcal{K}} \text{KL}\left( \text{softmax}\left( \frac{z_t^{(i)}}{T} \right) \parallel \text{softmax}\left( \frac{z_s^{(i)}}{T} \right) \right). \tag{4}$$

## 3 WHITTLE: A LIBRARY FOR SLM PRE-TRAINING AND DISTILLATION

Recent model releases such as LLaMA 3.1–8B, LLaMA 3.2–1B, and LLaMA 3.2–3B[2] leverage pruning and distillation to produce smaller variants, but their training recipes and code are closed-source, hindering reproducibility. Similarly, Minitron (Muralidharan et al., 2024) outlines best practices for SLM pretraining, but its implementation[3] is not readily generalizable across model families.

To address this gap, we present `whittle`, a fully open-source library that provides a reproducible, general-purpose pipeline for extracting and pretraining SLMs directly from Hugging Face models. `Whittle` supports a range of functionalities to allow for flexible search space design, sub-network search, extraction, pretraining and knowledge distillation. In this section, we outline the core functionalities of `whittle` and its API design:

**`set_sub_network()`.** Given a pretrained decoder-only LLM from `litgpt`, we first convert it into a `whittle` model to enable flexible sub-network extraction. To evaluate a sub-network using the `whittle` model, we dynamically activate only structured components of the LLM associated with that sub-network using the `set_sub_network()` API (Listing 4). It allows the user to explicitly set architectural parameters of the sub-network, such as embedding dimension, intermediate size, number of heads, layers, query groups, and head size, as well as indices for sampled neurons, layers, and heads. Importantly, it allows to vary the number of heads, head size, intermediate size, and query groups across layers. This function is a core utility in `whittle`, supporting downstream procedures such as search, pretraining, and distillation.

**`search()`.** The `search()` API (Listing 2, Appendix F) constructs a `whittle` super-network from a base HuggingFace model and facilitates automated sub-network selection. It supports evolutionary strategies as well as algorithms from `syne-tune` (Salinas et al., 2022)[4], and performs constrained search across parameter bins via rejection sampling. Each candidate sub-network is instantiated through `set_sub_network()` and evaluated on a task-specific metric, such as perplexity, to guide the search process.

**`convert_subnet_to_litgpt_model()`.** The `convert_subnet_to_litgpt_model()` function (Listing 3, Appendix F) transforms a selected sub-network configuration into a standalone GPT model within the `litgpt` framework. Given a super-network and a dictionary specifying architectural configurations (e.g., embedding dimension and number of heads), this utility extracts the corresponding sub-network and instantiates it as an independent GPT model. The resulting model can then be employed for downstream tasks such as pretraining, fine-tuning, or distillation.

**`pretrain()`.** The `pretrain()` function (Listing 1, Appendix F) enables pretraining of a sub-network initialized from a checkpointed GPT model. Given the model weights, a configuration file describing the sub-network architecture, and a target dataset, this utility restores the model and resumes training from the specified state.

**`distill()`.** The `distill()` function (Listing 5, Appendix F) supports knowledge distillation from a larger teacher model into a sub-network extracted from a checkpoint. Given a teacher model, sub-network configuration, and a target dataset, this utility trains the sub-network under the supervision of a specified teacher (e.g., `EleutherAI/pythia-12b`). Different distillation objectives (e.g., forward KL divergence) and constraints such as top-$k$ token selection are supported.

---

[2] `https://ai.meta.com/blog/llama-3-2-connect-2024-vision-edge-mobile-devices/`

[3] `https://github.com/NVIDIA-NeMo/NeMo/tree/main`

[4] `https://github.com/syne-tune/syne-tune`

Table 1: Search space configurations for different model families. Here, $e$ denotes embedding dimension, $h$ the number of attention heads, $h_s$ the head size, $l$ the number of layers and $d$ the MLP dimension.

| Base Model | $e$ | $h$ | $h_s$ | $l$ | $d$ |
|---|---|---|---|---|---|
| EleutherAI/pythia-6.9b | $[1, 4096]$ | $[1, 32]$ | $\{4, 6, 8, \ldots, 128\}$ | $[1, 32]$ | $[1, 16384]$ |
| EleutherAI/pythia-12b | $[1, 5120]$ | $[1, 40]$ | $\{4, 6, 8, \ldots, 128\}$ | $[1, 36]$ | $[1, 20480]$ |

## 4 EXPERIMENTS

Our study focuses on the Pythia (Biderman et al., 2023) family of models, which span sizes from 14M to 12B parameters; in particular, we use the 6.9B and 12B variants. Importantly, the modular design of our framework ensures that the methodology is readily applicable to any large language model supported by litgpt[5].

Our experiments are organized around three core components: (a) sub-network search, (b) pretraining of small language models (SLMs), and (c) distillation into SLMs. For each component, we outline the setup, present results, and highlight key insights. We now discuss them in turn.

### 4.1 SEARCH SPACE DEFINITIONS

Table 1 summarizes the search spaces for Pythia-6.9B and Pythia-12B, listing the allowable values for each transformer dimension. In the *coarse layer-wise* and *fine-grained layer-wise* settings, $h$, $h_s$, and $d$ are sampled independently at each layer. The *fine-grained* spaces extend this further with neuron-level sampling within each dimension, as detailed in Section 2.1. The size of each of the search spaces is listed in Table 16 (Appendix H).

### 4.2 EVOLUTIONARY SEARCH FOR OPTIMAL SLMS

**Search Setup** We apply Algorithm 1 (Appendix C.2) to conduct evolutionary search over parameter bins in Pythia-6.9B and Pythia-12B, considering the *coarse uniform*, *coarse layer-wise*, *fine-grained uniform*, and *fine-grained layer-wise*[6] search spaces from Section 2.1. We use perplexity on wikitext (Merity et al., 2017) as selection metric, with mutation and crossover probabilities fixed at $0.2$. For Pythia-6.9B and -12B, we define three bins with parameter counts of 385M–426M (bin-1), 961M–1.06B (bin-2), and 2.64B–2.91B (bin-3), centered at 5% of Pythia-410M, Pythia-1B, and Pythia-2.8B, respectively[7]. Each evolutionary run proceeds for 100 epochs, with results in the fine-grained layer-wise setting reported at the final epoch before rejection sampling becomes infeasible due to the combinatorial growth of the search space.

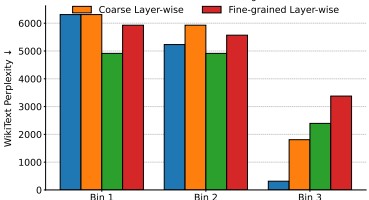

Figure 3: Best perplexity after evolutionary search based on perplexity for different search spaces.

**Results Discussion.** Figure 3 reports the perplexity of pruned sub-networks from Pythia-6.9B across bin-1, bin-2, and bin-3 under different search spaces on the wikitext test set. Note that these sub-networks are evaluated without any further pretraining or finetuning. We observe that searches constrained to the smaller *coarse uniform* and *coarse layer-wise* spaces generally yield more effective sub-networks.

---

[5]https://github.com/Lightning-AI/litgpt/

[6]In the fine-grained spaces, $h$, $h_s$, and $d$ are sampled independently at each layer, with additional neuron-level sampling within each dimension. See Section 2.1 for details.

[7]The bins were computed based on the *exact* number of parameters in the Pythia models

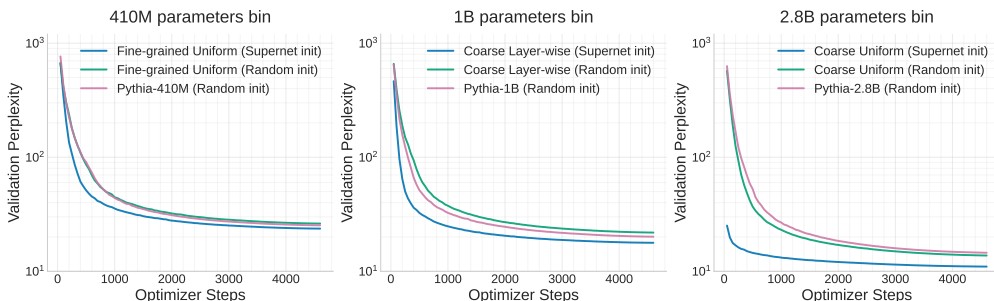

Figure 4: **Pretraining** Validation perplexity of the best sub-networks from each search space and the bin-center Pythia models (410M, 1B, 2.8B), all trained for 10B tokens with cross-entropy loss. Sub-networks are extracted from the Pythia-6.9B base model.

### 4.3 PRETRAINING OF SLMs

**Pretraining Setup.** We perform pretraining of our models on the **Nemotron-CC** dataset (Su et al., 2025). For each parameter bin and search space, we first conduct a set of low-fidelity experiments with a 2B-token budget to identify the most promising sub-network in each search space. Concretely, this involves evaluating the best candidate architecture for every bin across all four search spaces, resulting in $3\,(\text{bins}) \times 4\,(\text{search spaces}) = 12$ low-fidelity runs. We then select the top-ranked architecture from each bin (three architectures in total) and perform larger-scale pretraining with a 10B-token budget on **Nemotron-CC**. All models are trained with the standard next-token prediction objective using cross-entropy loss.

**Results and Discussion.** Figure 4 presents the pretraining results. We compare pretraining of the extracted best sub-network (*Supernet-init*) against two baselines: (i) *Random-init*, where the same architecture is trained with random initialization and a 10B-token budget, and (ii) the original Pythia model (center of the bin), also trained with random initialization and the same budget. Across parameter bins, initializing from the supernet yields consistent improvements in validation perplexity. Notably, the gains are most pronounced for `bin-3`, indicating that supernet initialization is particularly beneficial in higher-parameter regimes, where our model achieved the same validation perplexity with $\mathbf{5.16\times}$ fewer FLOP. Results for a token-budget of 100B can be found in Appendix J.

### 4.4 DISTILLATION OF SLMs

**Distillation Setup.** To further accelerate convergence, we distill knowledge from `Pythia-6.9B` and `Pythia-12B` teacher models. As described in Section 2, training is performed with a weighted combination of forward-KL divergence and cross-entropy loss (0.8 and 0.2, respectively). For computational efficiency, we apply top-$k$ logits distillation with $k = 1024$ and a distillation temperature of 0.9. For distillation, we select the best architectures from every bin, determined by pretraining for a small token budget of 2B tokens in Section 4.3, and train it with a larger token budget of 10B tokens with the distillation loss function on **Nemotron-CC**. When training a sub-network with distillation loss, we use the same model for the teacher as the one that the sub-network was extracted from (a sub-network extracted from `Pythia-6.9B` uses the `Pythia-6.9B` as the teacher model as well).

**Results and Discussion.** Figure 5 illustrates the effect of distillation. We find that distillation consistently improves perplexity in both `bin-1` and `bin-2`, with the model in `bin-2`. We also report the performance of distilled models on downstream tasks in Table 14 in Appendix D.

### 4.5 EVALUATION ON DOWNSTREAM TASKS

**Evaluation Setup.** We evaluate our pretrained and distilled sub-networks on different common-sense and question-answering type tasks. Specifically, we evaluate 0-shot performance on `copa`, `lambada_openai`, and `winogrande`, 5-shot performance on `MMLU`, and 10-shot performance on `arc-easy`, `arc_challenge`, `piqa`, and `hellaswag`. We report accuracy for `copa`, `lambada_openai`,

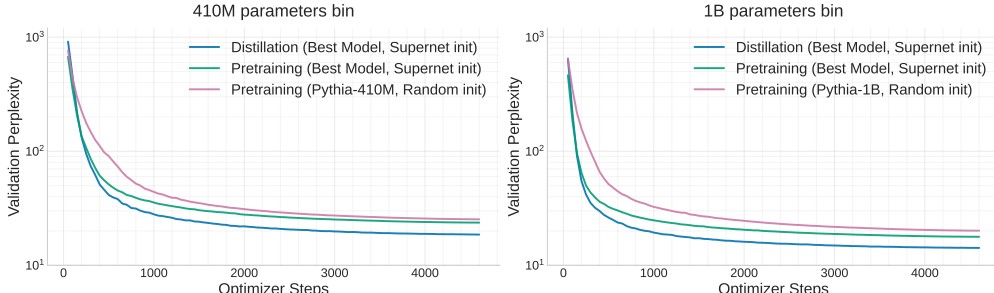

**Validation Perplexity Across Different Parameter Ranges (Pythia-6.9B as Source Model)**

Figure 5: **Distillation**: Comparison of validation perplexity for models trained with distillation loss v/s cross entropy loss. All sub-networks are extracted from Pythia-6.9B as a base model and trained for 10B tokens

| Base Model | Initialization | #Params | COPA | Lambda-OpenAI | Winogrande | MMLU | PIQA | ARC-challenge | ARC-easy | HellaSwag | Avg-acc | PPL-Nemotron-cc |
|---|---|---|---|---|---|---|---|---|---|---|---|---|
| Pythia-6.9B | Random Init | 389M | 59.00 | 18.51 | 51.14 | 26.43 | 63.87 | 24.74 | 46.80 | 31.28 | 37.77 | 26.20 |
| | Supernet Init | 389M | 61.00 | 24.02 | 51.54 | 26.35 | 65.34 | 24.57 | 51.38 | 33.11 | 39.33 | 23.66 |
| Pythia-12B | Random Init | 407M | 57.00 | 14.87 | 50.67 | 26.33 | 61.64 | 23.97 | 42.47 | 29.76 | 36.06 | 29.57 |
| | Supernet Init | 407M | 63.00 | 18.37 | 52.09 | 25.99 | 62.73 | 23.55 | 46.42 | 30.91 | 37.50 | 27.33 |
| Pythia-410M* | Random Init | 405M | 62.00 | 19.54 | 50.67 | 25.54 | 64.14 | 24.57 | 47.35 | 32.70 | 38.42 | 25.29 |
| Pythia-6.9B | Random Init | 1.04B | 64.00 | 23.36 | 51.54 | 26.62 | 65.78 | 27.13 | 51.80 | 36.83 | 40.22 | 21.84 |
| | Supernet Init | 1.04B | 66.00 | 38.52 | 51.46 | 26.09 | 69.26 | 30.12 | 63.51 | 45.20 | 43.74 | 17.77 |
| Pythia-12B | Random Init | 1.04B | 63.00 | 23.33 | 50.51 | 26.14 | 66.76 | 26.36 | 53.74 | 36.65 | 39.32 | 21.21 |
| | Supernet Init | 1.04B | 64.00 | 27.56 | 51.77 | 26.19 | 66.54 | 26.45 | 53.96 | 36.42 | 40.88 | 20.77 |
| Pythia-1B* | Random Init | 1.01B | 64.00 | 25.67 | 52.41 | 25.20 | 66.00 | 28.24 | 56.14 | 38.23 | 41.06 | 20.11 |
| Pythia-6.9B | Random Init | 2.91B | 61.00 | 26.49 | 52.10 | 26.39 | 67.74 | 28.33 | 57.83 | 41.12 | 41.31 | 13.75 |
| | Supernet Init | 2.91B | 66.00 | 50.16 | 56.91 | 26.45 | 72.69 | 33.87 | 67.09 | 53.40 | 47.41 | 10.99 |
| Pythia-12B | Random Init | 2.91B | 67.00 | 27.32 | 50.36 | 25.25 | 67.85 | 27.65 | 57.79 | 40.58 | 41.73 | 13.26 |
| | Supernet Init | 2.91B | 69.00 | 41.76 | 51.46 | 26.20 | 70.57 | 28.67 | 61.99 | 45.98 | 44.47 | 11.71 |
| Pythia-2.8B* | Random Init | 2.78B | 68.00 | 24.51 | 53.03 | 25.74 | 67.68 | 25.34 | 46.67 | 39.11 | 40.55 | 14.54 |

Table 2: Evaluation of sub-networks extracted from Pythia-6.9B and Pythia-12B after pretraining on 10B tokens. A Pythia model of comparable size is also trained on the same budget with random initialization to serve as a baseline (indicated with ∗). Reported numbers are metrics as defined in Section 4.5 (%).

`winogrande`, `MMLU` and length normalized accuracy for `piqa`, `arc_easy`, `arc_challenge` and `hellaswag`. We use lm-eval-harness[8] to perform evaluation on downstream tasks.

**Results Discussion.** Table 2 reports average downstream accuracies for our best sub-networks in each parameter bin pretrained with a 10B-token budget. For comparison, we also include Pythia-410M, 1B, and 2.8B models trained with the same budget. Across all bins, `Supernet-init` outperforms both `Random-init` (for the same extracted architecture) and the original Pythia architectures (bin centers). Furthermore, sub-networks extracted from the smaller base model (Pythia-6.9B) consistently outperform those from the larger base (Pythia-12B). We present results of our distilled models on downstream tasks in Table 14.

## 5 ABLATIONS

In this section, we conduct ablation studies to examine the effect of four key factors in our framework: (a) the choice of search space, (b) the loss function used for distillation, (c) the performance metric employed during search.

**Granularity of Search Spaces.** Figure 6 illustrates the effect of varying search space granularity. We find that different bins benefit from distinct choices: for `bin-1`, *fine-grained uniform* search space is optimal; for `bin-2`, *coarse layer-wise* performs best; and for `bin-3`, *coarse uniform* yields the strongest results.

**Full logits vs. top-$k$ logits.** In our distillation experiments in Section 4, following (Team et al., 2025), we use top-k logit based distillation. Here, we ablate this choice for the distillation loss by

---

[8]https://github.com/EleutherAI/lm-evaluation-harness

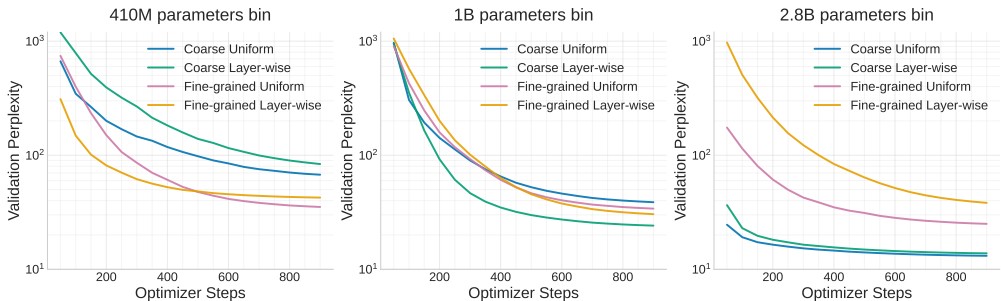

Figure 6: Validation perplexity of the best models from each search space found via evolutionary search. All models are initialized with Pythia-6.9B weights and trained for 2 billion tokens. Within each bin, the models' parameter counts fall within a $\pm 5\%$ range of that bin's target size.

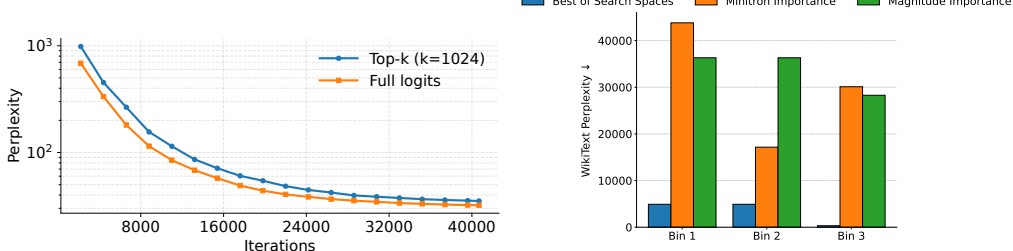

Figure 7: Full vs. top-$k$ logit distillation.

Figure 8: Comparison of search guided by importance metrics and perplexity. We report results in the best search space for each bin.

comparing supervision from the full teacher distribution against a truncated variant using only the top-$k$ logits (Figure 7). This isolates how much of the teacher's probability mass is required for effective transfer. We find that, in general, *distilling from the full-logit distribution yields a lower perplexity*.

**Metric for Searching sub-networks.**   Finally, in Figure 8, we evaluate different search metrics. Specifically, we compare activation-based importance scores (as in Minitron (Muralidharan et al., 2024)) and weight-magnitude scores (Han et al., 2015) against directly optimizing for perplexity in our setup. We define the details of the importance score computation procedure, i.e. the metric guiding the search, in Appendix D. All searches are run with for 100 epochs. We find that perplexity-based search consistently achieves lower perplexity than proxy metrics, suggesting that importance and magnitude scores are less reliable indicators of sub-network quality.

## 6   CONCLUSION

We present a principled framework for initializing small language models (SLMs) by extracting sub-networks from a larger pre-trained teacher network. Our experiments demonstrate that this approach accelerates the overall pre-training process of SLMs by up to $9.2\times$ compared to baseline SLM models of similar size. To select the sub-network, we employ a constrained evolutionary search strategy that identifies optimal candidates based on validation performance. Further, we analyze four different search spaces of increasing granularity and demonstrate that for the larger variants of SLMs, the least granular search space (*coarse uniform*) yields the best model. The smaller variants, however, benefit from more granular search spaces such as *fine-grained uniform* and *coarse layer-wise*.

For future work, we aim to derive scaling laws to better understand the impact of improved initialization strategies as model and data scales increase. Additionally, we plan to investigate the effect of teacher model choice on student performance, particularly in domain-specific settings. For example, it remains an open question whether a multilingual teacher provides advantages over an English-only teacher when training a monolingual student model.

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

# A   RELATED WORK

**Model Pruning.**   Pruning is a core approach for compressing neural networks by removing redundant parameters while preserving accuracy. Early work on unstructured magnitude pruning (LeCun et al., 1990; Han et al., 2016) achieved high sparsity with minimal accuracy loss, but offered limited inference benefits on modern hardware. This motivated structured and semi-structured pruning methods that remove neurons, filters, or enforce hardware-friendly sparsity patterns (Li et al., 2017; Zhou et al., 2021; Ma et al., 2023; Frantar and Alistarh, 2023). The Lottery Ticket Hypothesis (LTH) (Frankle and Carbin, 2018) provided a compelling rationale, showing that large networks contain sub-networks ("winning tickets") that can train in isolation to match full-model performance. Subsequent work examined their generalization across architectures and optimizers (Morcos et al., 2019; Desai et al., 2019), their stabilization and theoretical underpinnings (Frankle et al., 2019; Malach et al., 2020), and their presence in large pretrained language models (Chen et al., 2020; Prasanna et al., 2020; Liang et al., 2021). These advances highlight pruning as a powerful tool for efficient deployment in resource-constrained settings. Central to both pruning and ticket discovery is the design of *importance scores*—criteria based on weight magnitude, gradients, or activations (Molchanov et al., 2019; Frantar and Alistarh, 2023; An et al., 2024) that estimate which components can be removed with minimal loss. However, efficiently scaling such methods to billion-parameter LMs remains a major challenge. Our work addresses this gap by introducing a framework for discovering high-quality sub-networks that is *efficient, scalable, and easily parallelizable*.

**Comparison with Sheared LLaMA and DRPruning**   Sheared LLaMA (Xia et al.) frames pruning as a constrained optimization problem, updating weights and masks together via repeated pretraining-like steps. It also uses dynamic batch loading, which blurs the distinction between pruning benefits and training effects, while adding considerable computational overhead. Similarly, DRPruning (Deng et al., 2025) learns structured masks under the full pretraining objective and further introduces distributionally robust data reweighting, conflating pruning benefits with data-selection effects. In contrast, `whittle` identifies strong SLM initializations within a target parameter range using only preplexity estimates from a forward-pass, avoiding mask/weight optimization, additional training heuristics, and data reweighting. This yields a simple approach that fits cleanly into standard next-token-prediction pretraining pipelines.

**Knowledge Distillation (KD).**   KD compresses large language models by transferring knowledge from a teacher to a smaller student, aiming to preserve accuracy while reducing compute (Hinton et al., 2015; Xu et al., 2024). For autoregressive LMs, this is typically done in two ways. *Logit-based distillation* trains the student to match the teacher's output distribution via KL-divergence, often with top-$k$ or top-$p$ truncation to mitigate noise from heavy-tailed distributions (Hinton et al., 2015; Kim and Rush, 2016; Sanh et al., 2019; Team et al., 2024). *Representation-based distillation* instead aligns internal dynamics, training the student to mimic hidden states or their projections using MSE losses (Romero et al., 2015; Jiao et al., 2020; Wang et al., 2020b). These complementary strategies highlight KD's versatility in shaping both outputs and internal representations. Beyond compression, KD smooths decision boundaries and provides richer training signals, often yielding faster and more stable convergence. Building on these insights, we demonstrate the effectiveness of KD as a key ingredient for efficient SLM pretraining.

**Neural Architecture Search (NAS).**   NAS (White et al., 2023; Elsken et al., 2019b) automates the exploration of large architecture spaces. Existing approaches include black-box optimization (Zoph and Le, 2017; White et al., 2021; Real et al., 2019; Shen et al., 2023; Zhou et al., 2019; Schrodi et al., 2023), which repeatedly train and evaluate candidates, and gradient-based methods (Liu et al., 2019; Dong and Yang, 2019; Chen et al., 2021; Zela et al., 2020), which perform differentiable search over a weight-sharing supernetwork. Extensions incorporate hardware-awareness and multi-objective criteria (Sukthanker et al., 2025; Elsken et al., 2019a; Lu et al., 2019; Hsu et al., 2018; Lu et al., 2020; Sukthanker et al., 2024; Lee et al., 2021; Li et al., 2021; Klein et al., 2024), jointly optimizing accuracy, efficiency, and deployment constraints. A major limitation, however, is the expensive supernet pretraining required by most methods (Cai et al., 2020; Sukthanker et al., 2024; Wang et al., 2020a), which is prohibitive at the scale of LLMs. Our approach sidesteps this by leveraging open-source pretrained LLMs as the basis for search, eliminating supernet pretraining. Moreover, unlike

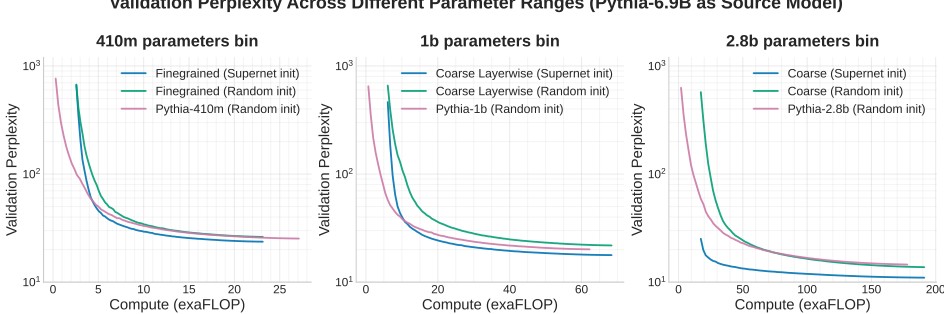

Figure 9: Validation perplexity across different parameter ranges (offset with search cost).

traditional NAS that seeks architectures for direct deployment, we focus on discovering sub-networks that provide strong initializations for efficient pretraining.

**SLM Pretraining in Practice.** Recent open-source releases often provide families of models ranging from compact Small Language Models (SLMs) to much larger variants. SLMs are especially important for edge deployment, where efficiency and memory are critical. A straightforward way to obtain them is to train models across multiple scales (Biderman et al., 2023), but this is computationally costly. To reduce training demands, recent work instead trains a large base model and extracts smaller ones via pruning and distillation (Muralidharan et al., 2024; Meta AI, 2024; Team et al., 2025), or relies solely on distillation from a larger teacher, as in Gemma-3 (Team et al., 2025). Despite this progress, there remains no principled framework for compute-efficient SLM pretraining. Our work addresses this gap through a systematic study of sub-network extraction and initialization strategies, combined with pipeline designs and loss functions for training high-performing SLMs.

# B ADDITIONAL EXPERIMENTAL DETAILS

## B.1 HYPERPARAMETER CONFIGURATIONS OF EXPERIMENTS

In Tables 4 - 8, we present the hyperparameter settings for all our experiments.

## B.2 COMPUTATIONAL COST OF THE EVOLUTIONARY SEARCH

In this section, we provide an overview of the cost overhead introduced by evolutionary search. In each bin, for every search space, we sample and evaluate a total of 5,050 subnetworks during the evolutionary search. For each candidate model, we computed the perplexity on 1,000 sequences of length 512. We approximate the average FLOP of the models in bins 1, 2 and 3 as the FLOPs of Pythia-410M, Pythia-1B, and Pythia-2.8B, since these models serve as the center of the bins. The total computational cost of the search for each search space is reported in Table 9. For comparison, the cost of pretraining the best model found in each bin on 10B tokens is as presented in Table 10. As Tables 9 and 10 indicate, the search phase consumes only a small fraction of the overall pretraining budget.

**Revised Cost Savings.** We include the cost of the evolutionary search when computing the total cost savings achieved by our method. The updated FLOP-savings factors are reported below in Table 11. Additionally, we include the FLOP-savings factor considering *only* the pretraining budget in Table 12.

Furthermore, in Figure 9, we present the validation perplexity across different parameter ranges, taking the search cost into account.

# C ADDITIONAL METHODOLOGICAL DETAILS

Table 3: Hyperparameters used for the Best Performing Subnets per Bin (Parameter Range) from the Pythia-12B Model

| Search Space | Parameter range | Hyperparameter Type | | Value |
|---|---|---|---|---|
| Evolutionary Search Coarse | Bin 1 385M–426M | Model & Data | Model Name | pythia-12b |
| | | | Precision | bf16-mixed |
| | | | Dataset | Nemotron-CC |
| | | Optimizer & Regularization | Optimizer | AdamW |
| | | | Learning Rate | $3 \times 10^{-4}$ |
| | | | Min Learning Rate | $3 \times 10^{-5}$ |
| | | | Weight Decay | 0.01 |
| | | | AdamW $\beta_1, \beta_2$ | 0.9, 0.95 |
| | | | Gradient Clipping Norm | 1.0 |
| | | Training & Batching | Total Training Tokens | 50B |
| | | | Global Batch Size | 1056 |
| | | | Micro Batch Size | 8 |
| | | | LR Warmup Steps | 0 |
| | | | Max Sequence Length | 2048 |
| | | | Seed | 42 |
| Evolutionary Search Coarse Layerwise | Bin 2 961M–1.06B | Model & Data | Model Name | pythia-12b |
| | | | Precision | bf16-mixed |
| | | | Dataset | Nemotron-CC |
| | | Optimizer & Regularization | Optimizer | AdamW |
| | | | Learning Rate | $3 \times 10^{-4}$ |
| | | | Min Learning Rate | $3 \times 10^{-5}$ |
| | | | Weight Decay | 0.01 |
| | | | AdamW $\beta_1, \beta_2$ | 0.9, 0.95 |
| | | | Gradient Clipping Norm | 1.0 |
| | | Training & Batching | Total Training Tokens | 50B |
| | | | Global Batch Size | 1056 |
| | | | Micro Batch Size | 8 |
| | | | LR Warmup Steps | 0 |
| | | | Max Sequence Length | 2048 |
| | | | Seed | 42 |
| Evolutionary Search Coarse | Bin 3 2.64B–2.91B | Model & Data | Model Name | pythia-12b |
| | | | Precision | bf16-mixed |
| | | | Dataset | Nemotron-CC |
| | | Optimizer & Regularization | Optimizer | AdamW |
| | | | Learning Rate | $1.6 \times 10^{-4}$ |
| | | | Min Learning Rate | $1.6 \times 10^{-5}$ |
| | | | Weight Decay | 0.01 |
| | | | AdamW $\beta_1, \beta_2$ | 0.9, 0.95 |
| | | | Gradient Clipping Norm | 1.0 |
| | | Training & Batching | Total Training Tokens | 50B |
| | | | Global Batch Size | 1056 |
| | | | Micro Batch Size | 16 |
| | | | LR Warmup Steps | 238 |
| | | | Max Sequence Length | 2048 |
| | | | Seed | 42 |

Table 4: Hyperparameters used for the Best Performing Subnets per Bin (Parameter Range) from the Pythia-6.9B Model

| Search Space | Parameter range | Hyperparameter Type | | Value |
|---|---|---|---|---|
| Evolutionary Search Finegrained | Bin 1 385M–426M | Model & Data | Model Name | pythia-6.9b |
| | | | Precision | bf16-mixed |
| | | | Dataset | Nemotron-CC |
| | | Optimizer & Regularization | Optimizer | AdamW |
| | | | Learning Rate | $3 \times 10^{-4}$ |
| | | | Min Learning Rate | $3 \times 10^{-5}$ |
| | | | Weight Decay | 0.01 |
| | | | AdamW $\beta_1, \beta_2$ | 0.9, 0.95 |
| | | | Gradient Clipping Norm | 1.0 |
| | | Training & Batching | Total Training Tokens | 50B |
| | | | Global Batch Size | 1056 |
| | | | Micro Batch Size | 6 |
| | | | LR Warmup Steps | 0 |
| | | | Max Sequence Length | 2048 |
| | | | Seed | 42 |
| Evolutionary Search Coarse Layerwise | Bin 2 961M–1.06B | Model & Data | Model Name | pythia-6.9b |
| | | | Precision | bf16-mixed |
| | | | Dataset | Nemotron-CC |
| | | Optimizer & Regularization | Optimizer | AdamW |
| | | | Learning Rate | $3 \times 10^{-4}$ |
| | | | Min Learning Rate | $3 \times 10^{-5}$ |
| | | | Weight Decay | 0.01 |
| | | | AdamW $\beta_1, \beta_2$ | 0.9, 0.95 |
| | | | Gradient Clipping Norm | 1.0 |
| | | Training & Batching | Total Training Tokens | 50B |
| | | | Global Batch Size | 1056 |
| | | | Micro Batch Size | 4 |
| | | | LR Warmup Steps | 0 |
| | | | Max Sequence Length | 2048 |
| | | | Seed | 42 |
| Evolutionary Search Coarse | Bin 3 2.64B–2.91B | Model & Data | Model Name | pythia-6.9b |
| | | | Precision | bf16-mixed |
| | | | Dataset | Nemotron-CC |
| | | Optimizer & Regularization | Optimizer | AdamW |
| | | | Learning Rate | $1.6 \times 10^{-4}$ |
| | | | Min Learning Rate | $1.6 \times 10^{-5}$ |
| | | | Weight Decay | 0.01 |
| | | | AdamW $\beta_1, \beta_2$ | 0.9, 0.95 |
| | | | Gradient Clipping Norm | 1.0 |
| | | Training & Batching | Total Training Tokens | 50B |
| | | | Global Batch Size | 1056 |
| | | | Micro Batch Size | 16 |
| | | | LR Warmup Steps | 238 |
| | | | Max Sequence Length | 2048 |
| | | | Seed | 42 |

Table 5: Hyperparameters used for Distillation Experiments on the Best Performing Subnets per Bin (Parameter Range) from the Pythia-12B Model

| Search Space | Parameter range | Hyperparameter Type | | Value |
|---|---|---|---|---|
| Evolutionary Search Coarse | Bin 1 385M–426M | Model & Data | Teacher Model Precision Dataset | pythia-12b bf16-mixed Nemotron-CC |
| | | Optimizer & Regularization | Optimizer Learning Rate Min Learning Rate Weight Decay AdamW $\beta_1, \beta_2$ Gradient Clipping Norm | AdamW $3 \times 10^{-4}$ $3 \times 10^{-5}$ 0.01 $0.9, 0.95$ 1.0 |
| | | Distillation | $\alpha$ $\beta$ Temperature Logits | 0.2 0.8 0.9 Top-1024 |
| | | Training & Batching | Total Training Tokens Global Batch Size Micro Batch Size LR Warmup Steps Max Sequence Length Seed | 10B 1056 2 0 2048 42 |
| Evolutionary Search Coarse Layerwise | Bin 2 961M–1.06B | Model & Data | Teacher Model Precision Dataset | pythia-12b bf16-true Nemotron-CC |
| | | Optimizer & Regularization | Optimizer Learning Rate Min Learning Rate Weight Decay AdamW $\beta_1, \beta_2$ Gradient Clipping Norm | AdamW $3 \times 10^{-4}$ $3 \times 10^{-5}$ 0.01 $0.9, 0.95$ 1.0 |
| | | Distillation | $\alpha$ $\beta$ Temperature Logits | 0.2 0.8 0.9 Top-1024 |
| | | Training & Batching | Total Training Tokens Global Batch Size Micro Batch Size LR Warmup Steps Max Sequence Length Seed | 10B 1056 8 0 2048 42 |
| Evolutionary Search Coarse | Bin 3 2.64B–2.91B | Model & Data | Teacher Model Precision Dataset | pythia-12b bf16-true Nemotron-CC |
| | | Optimizer & Regularization | Optimizer Learning Rate Min Learning Rate Weight Decay AdamW $\beta_1, \beta_2$ Gradient Clipping Norm | AdamW $1.6 \times 10^{-4}$ $1.6 \times 10^{-5}$ 0.01 $0.9, 0.95$ 1.0 |
| | | Distillation | $\alpha$ $\beta$ Temperature Logits | 0.2 0.8 0.9 Top-1024 |
| | | Training & Batching | Total Training Tokens Global Batch Size Micro Batch Size LR Warmup Steps Max Sequence Length Seed | 10B 1056 4 238 2048 42 |

Table 6: Hyperparameters used for Distillation Experiments on the Best Performing Subnets per Bin (Parameter Range) from the Pythia-6.9B Model

| Search Space | Parameter range | Hyperparameter Type | | Value |
|---|---|---|---|---|
| Evolutionary Search Finegrained | Bin 1 385M–426M | Model & Data | Teacher Model | pythia-6.9b |
| | | | Precision | bf16-mixed |
| | | | Dataset | Nemotron-CC |
| | | Optimizer & Regularization | Optimizer | AdamW |
| | | | Learning Rate | $3 \times 10^{-4}$ |
| | | | Min Learning Rate | $3 \times 10^{-5}$ |
| | | | Weight Decay | 0.01 |
| | | | AdamW $\beta_1, \beta_2$ | 0.9, 0.95 |
| | | | Gradient Clipping Norm | 1.0 |
| | | Distillation | $\alpha$ | 0.2 |
| | | | $\beta$ | 0.8 |
| | | | Temperature | 0.9 |
| | | | Logits | Top-1024 |
| | | Training & Batching | Total Training Tokens | 10B |
| | | | Global Batch Size | 1056 |
| | | | Micro Batch Size | 6 |
| | | | LR Warmup Steps | 0 |
| | | | Max Sequence Length | 2048 |
| | | | Seed | 42 |
| Evolutionary Search Coarse Layerwise | Bin 2 961M–1.06B | Model & Data | Teacher Model | pythia-6.9b |
| | | | Precision | bf16-mixed |
| | | | Dataset | Nemotron-CC |
| | | Optimizer & Regularization | Optimizer | AdamW |
| | | | Learning Rate | $3 \times 10^{-4}$ |
| | | | Min Learning Rate | $3 \times 10^{-5}$ |
| | | | Weight Decay | 0.01 |
| | | | AdamW $\beta_1, \beta_2$ | 0.9, 0.95 |
| | | | Gradient Clipping Norm | 1.0 |
| | | Distillation | $\alpha$ | 0.2 |
| | | | $\beta$ | 0.8 |
| | | | Temperature | 0.9 |
| | | | Logits | Top-1024 |
| | | Training & Batching | Total Training Tokens | 10B |
| | | | Global Batch Size | 1056 |
| | | | Micro Batch Size | 4 |
| | | | LR Warmup Steps | 0 |
| | | | Max Sequence Length | 2048 |
| | | | Seed | 42 |
| Evolutionary Search Coarse | Bin 3 2.64B–2.91B | Model & Data | Teacher Model | pythia-6.9b |
| | | | Precision | bf16-mixed |
| | | | Dataset | Nemotron-CC |
| | | Optimizer & Regularization | Optimizer | AdamW |
| | | | Learning Rate | $1.6 \times 10^{-4}$ |
| | | | Min Learning Rate | $1.6 \times 10^{-5}$ |
| | | | Weight Decay | 0.01 |
| | | | AdamW $\beta_1, \beta_2$ | 0.9, 0.95 |
| | | | Gradient Clipping Norm | 1.0 |
| | | Distillation | $\alpha$ | 0.2 |
| | | | $\beta$ | 0.8 |
| | | | Temperature | 0.9 |
| | | | Logits | Top-1024 |
| | | Training & Batching | Total Training Tokens | 10B |
| | | | Global Batch Size | 1056 |
| | | | Micro Batch Size | 4 |
| | | | LR Warmup Steps | 238 |
| | | | Max Sequence Length | 2048 |
| | | | Seed | 42 |

Table 7: Hyperparameters used for Distillation Ablation Experiments on a Subnet from the Pythia-6.9B Model using varying Teacher Model Sizes

| Search Space | Parameter range | Hyperparameter Type | | Value |
|---|---|---|---|---|
| Evolutionary Search Finegrained | Bin 1 385M–426M | Model & Data | Teacher Model | pythia-1b |
| | | | Precision | bf16-mixed |
| | | | Dataset | Nemotron-CC |
| | | Optimizer & Regularization | Optimizer | AdamW |
| | | | Learning Rate | $3 \times 10^{-4}$ |
| | | | Min Learning Rate | $3 \times 10^{-5}$ |
| | | | Weight Decay | 0.01 |
| | | | AdamW $\beta_1, \beta_2$ | 0.9, 0.95 |
| | | | Gradient Clipping Norm | 1.0 |
| | | Distillation | $\alpha$ | 0.2 |
| | | | $\beta$ | 0.8 |
| | | | Temperature | 0.9 |
| | | | Logits | Top-1024 |
| | | Training & Batching | Total Training Tokens | 2B |
| | | | Global Batch Size | 1056 |
| | | | Micro Batch Size | 6 |
| | | | LR Warmup Steps | 0 |
| | | | Max Sequence Length | 2048 |
| | | | Seed | 42 |
| Evolutionary Search Finegrained | Bin 1 385M–426M | Model & Data | Teacher Model | pythia-6.9b |
| | | | Precision | bf16-mixed |
| | | | Dataset | Nemotron-CC |
| | | Optimizer & Regularization | Optimizer | AdamW |
| | | | Learning Rate | $3 \times 10^{-4}$ |
| | | | Min Learning Rate | $3 \times 10^{-5}$ |
| | | | Weight Decay | 0.01 |
| | | | AdamW $\beta_1, \beta_2$ | 0.9, 0.95 |
| | | | Gradient Clipping Norm | 1.0 |
| | | Distillation | $\alpha$ | 0.2 |
| | | | $\beta$ | 0.8 |
| | | | Temperature | 0.9 |
| | | | Logits | Top-1024 |
| | | Training & Batching | Total Training Tokens | 2B |
| | | | Global Batch Size | 1056 |
| | | | Micro Batch Size | 6 |
| | | | LR Warmup Steps | 0 |
| | | | Max Sequence Length | 2048 |
| | | | Seed | 42 |

Table 8: Hyperparameters used for Distillation Ablation Experiments on a Subnet from the Pythia-6.9B Model — Top-K vs. Full Logits

| Search Space | Parameter range | Hyperparameter Type | | Value |
|---|---|---|---|---|
| | | Model & Data | Teacher Model | pythia-6.9b |
| | | | Precision | bf16-mixed |
| | | | Dataset | Nemotron-CC |
| | | Optimizer & Regularization | Optimizer | AdamW |
| | | | Learning Rate | $3 \times 10^{-4}$ |
| | | | Min Learning Rate | $3 \times 10^{-5}$ |
| | | | Weight Decay | 0.01 |
| | | | AdamW $\beta_1, \beta_2$ | 0.9, 0.95 |
| Evolutionary Search Finegrained | Bin 1 385M–426M | | Gradient Clipping Norm | 1.0 |
| | | Distillation | $\alpha$ | 0.2 |
| | | | $\beta$ | 0.8 |
| | | | Temperature | 0.9 |
| | | | Logits | Top-1024 |
| | | Training & Batching | Total Training Tokens | 2B |
| | | | Global Batch Size | 1056 |
| | | | Micro Batch Size | 6 |
| | | | LR Warmup Steps | 0 |
| | | | Max Sequence Length | 2048 |
| | | | Seed | 42 |
| | | Model & Data | Teacher Model | pythia-6.9b |
| | | | Precision | bf16-mixed |
| | | | Dataset | Nemotron-CC |
| | | Optimizer & Regularization | Optimizer | AdamW |
| | | | Learning Rate | $3 \times 10^{-4}$ |
| | | | Min Learning Rate | $3 \times 10^{-5}$ |
| | | | Weight Decay | 0.01 |
| | | | AdamW $\beta_1, \beta_2$ | 0.9, 0.95 |
| Evolutionary Search Finegrained | Bin 1 385M–426M | | Gradient Clipping Norm | 1.0 |
| | | Distillation | $\alpha$ | 0.2 |
| | | | $\beta$ | 0.8 |
| | | | Temperature | 0.9 |
| | | | Logits | Full |
| | | Training & Batching | Total Training Tokens | 2B |
| | | | Global Batch Size | 1056 |
| | | | Micro Batch Size | 6 |
| | | | LR Warmup Steps | 0 |
| | | | Max Sequence Length | 2048 |
| | | | Seed | 42 |

| Bin | Search Cost (exaFLOP) |
|-----|-----------------------|
| 1 | 2.3 |
| 2 | 5.4 |
| 3 | 15.4 |

Table 9: Cost of Evolutionary Search for different bins

| Bin | Pretraining Cost (exaFLOP) |
|-----|----------------------------|
| 1 | 20.9 |
| 2 | 63.3 |
| 3 | 176.6 |

Table 10: Cost of pretraining the best models in different bins

| Bin | FLOP Savings Factor |
|-----|---------------------|
| 1 | 1.71× |
| 2 | 1.75× |
| 3 | 5.16× |

Table 11: FLOP-saving factors for all bins compared to pretraining the corresponding Pythia architectures.

| Bin | FLOP Savings Factor (excluding search cost) |
|-----|---------------------------------------------|
| 1 | 2.0× |
| 2 | 2.07× |
| 3 | 9.32× |

Table 12: FLOP-saving factors for all bins compared to pretraining the corresponding Pythia architectures (without considering the search cost).

## C.1 ATTENTION MASKING

The attention mechanism used in transformer blocks naturally supports sub-network extraction. In practice, this means that an attention mechanism can be masked to yield a smaller, distinct type of attention. Figure 11 provides an overview of the main variants—multi-head attention (MHA), multi-query attention (MQA), and grouped-query attention (GQA). Since GQA serves as a superclass of these mechanisms, it can be transformed into either MHA or MQA. An illustration of this transformation is shown in Figure 12.

## C.2 EVOLUTIONARY SEARCH ALGORITHM

We present the details of our evolutionary search algorithm in Algorithm 1.

## D ADDITIONAL RESULTS

Below, we present additional experimental results. Figure 13 shows the effect of using different teachers for knowledge distillation, Figure 14 shows the evolutionary search trajectory for different parameter bins, with the best perplexity marked in red. Table 14, presents the result of distilled models on downstream tasks. Table 17, provides the results on an extended set of common sense reasoning based downstream tasks.

Figures 15–17 summarize the training behavior of the best models across different settings. The results highlight how architectures extracted from different search spaces (Figure 15), weight initialization strategies (Figure 16), and the use of distillation (Figure 17) affect convergence and final performance.

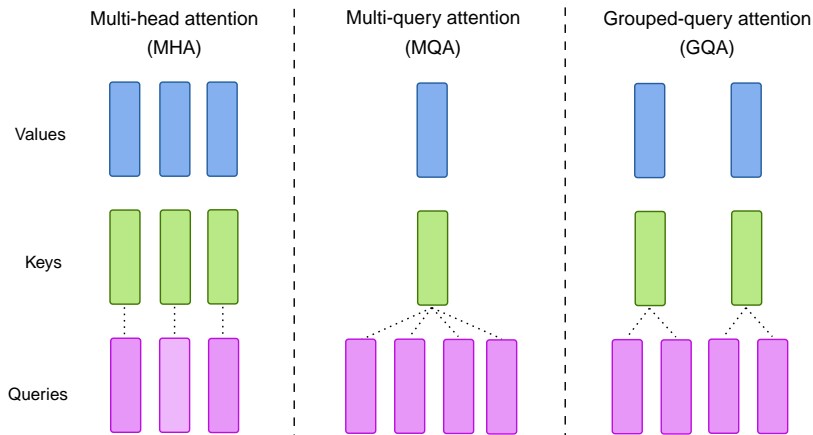

Figure 10: An overview of the Whittle library.

Figure 11: An illustration of the different types of attention mechanisms. In multi-head attention (MHA), each query is paired with its own key and value; in multi-query attention (MQA), multiple queries share a single key–value pair; and in grouped-query attention (GQA), multiple key–value pairs are used, with each pair serving more than one query.

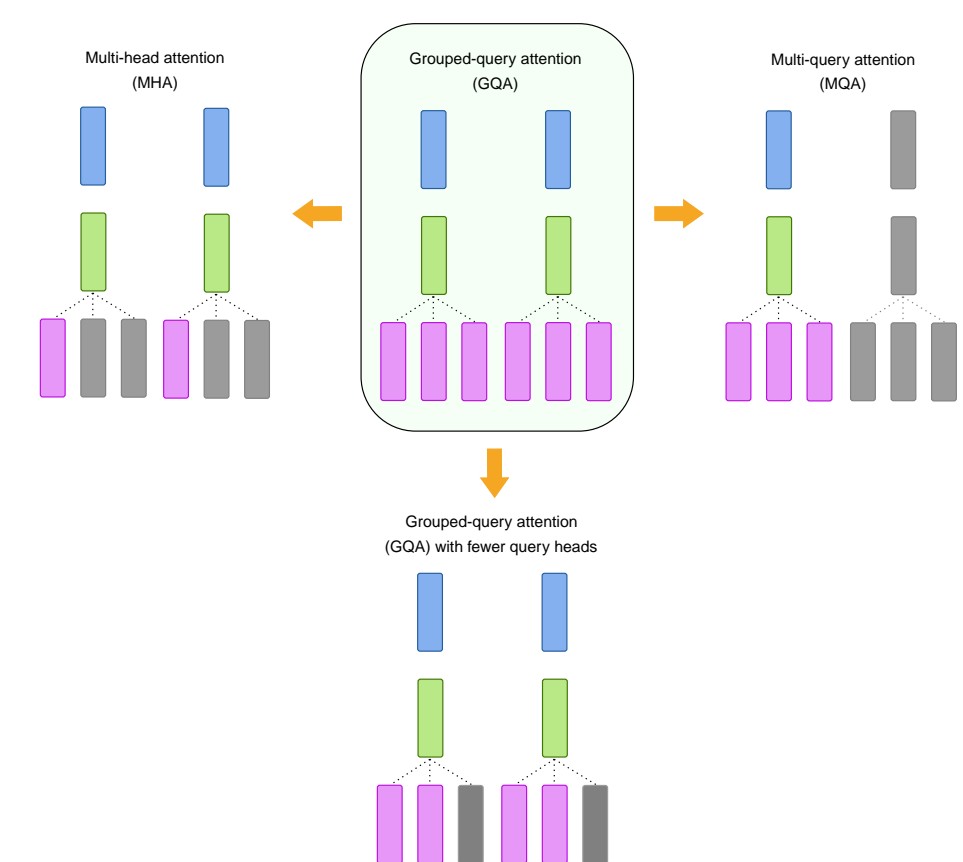

Figure 12: An example of how grouped-query attention (GQA) can be masked to emulate other forms of attention, such as multi-head or multi-query attention. The masked heads are shown in gray. Note that GQA can also be reduced to fewer query heads while preserving the same number of groups.

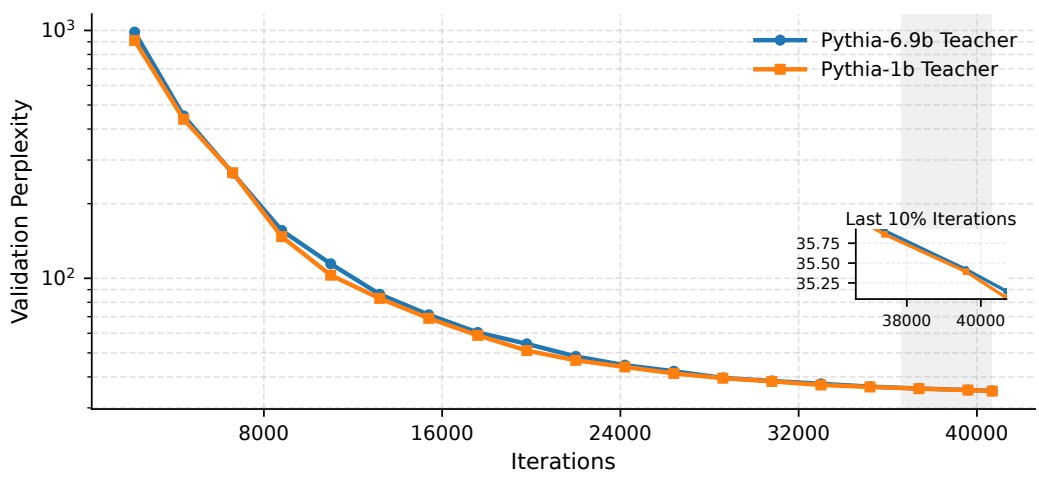

Figure 13: Teacher size vs. student performance with a 2B-token budget. A Pythia-1B teacher achieves validation perplexity 35.06, slightly better than Pythia-6.9B (35.15).

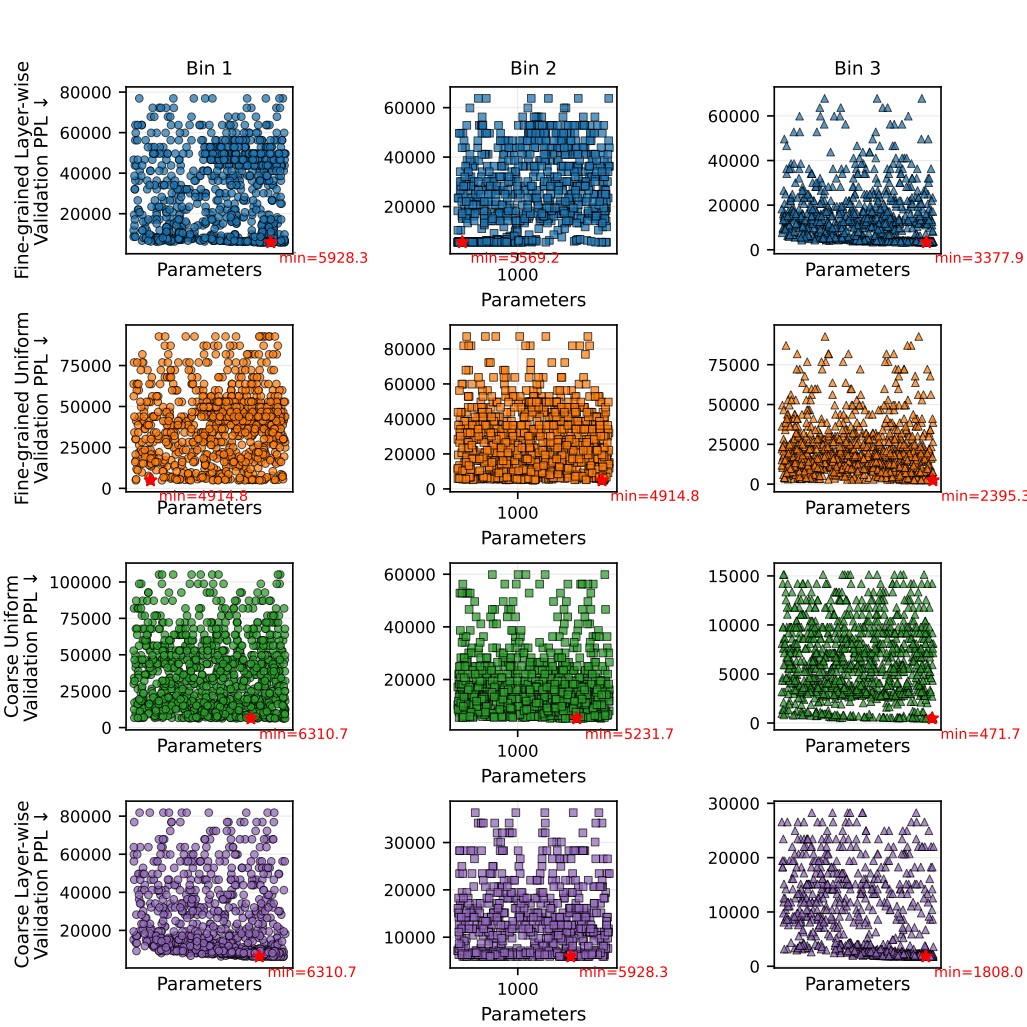

Figure 14: Evolutionary search on *coarse uniform*, *coarse layer-wise*, *fine-grained uniform* and *fine-grained layer-wise* search spaces for Pythia-6.9B. Minimum perplexity for each bin marked in red

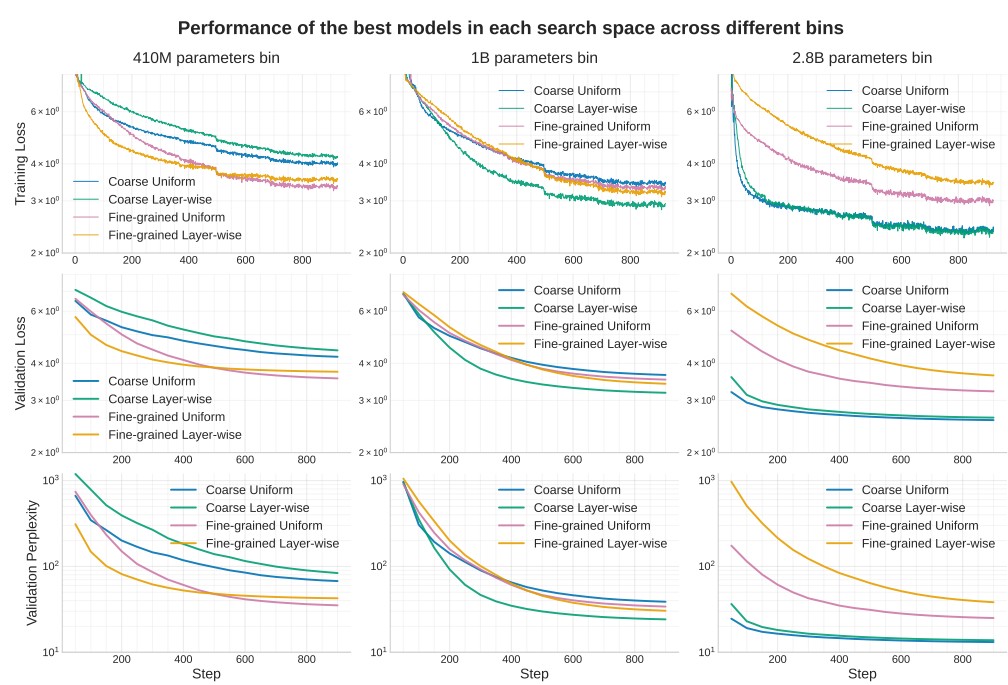

Figure 15: Training curves of the best models from each search space extracted from Pythia-6.9b (trained for 2 billion tokens)

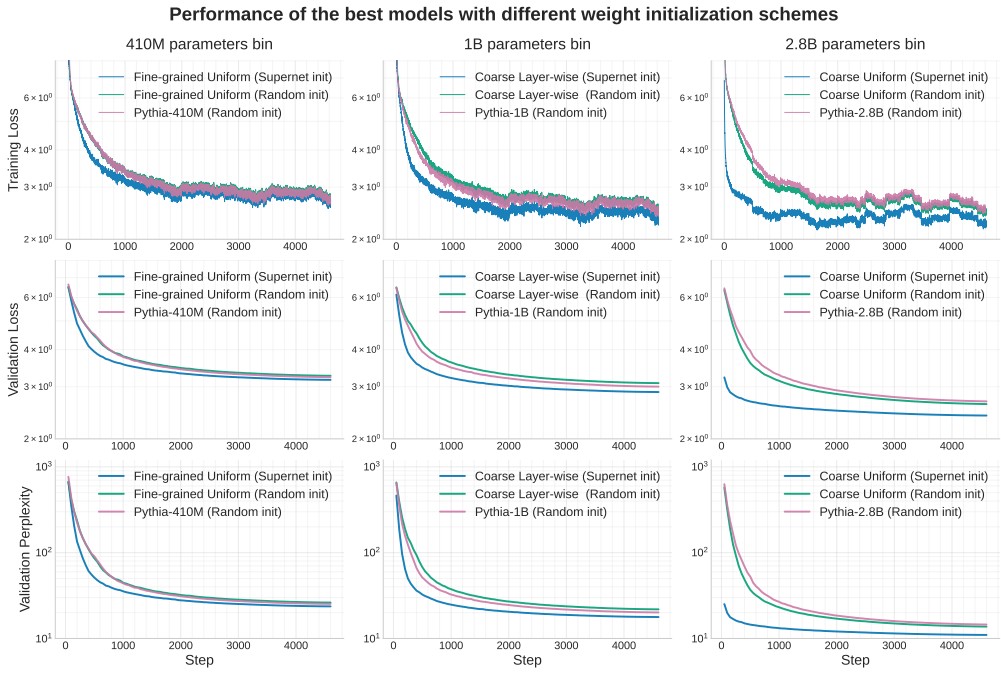

Figure 16: Training curves of the best models found in each bin, initialized with supernet weights as well as random weights. A Pythia model of comparable size is also trained with random initialization in each bin as a baseline. The models are trained with 10 billion tokens.

| Base Model | Initialization | #Params | COPA | OpenBookQA | Lambada-OpenAI | Winogrande | Social IQA | MMLU-cont. | MMLU | CommonsenseQA | PIQA | ARC-challenge | ARC-easy | HellaSwag | BoolQ | Avg-acc | PPL-Nemotron-cc |
|---|---|---|---|---|---|---|---|---|---|---|---|---|---|---|---|---|---|
| Pythia-6.9B | Random Init | 389M | 59.00 | 29.20 | 18.51 | 51.14 | 36.59 | 25.82 | 26.43 | 19.41 | 63.87 | 24.74 | 46.80 | 31.28 | 58.17 | 37.77 | 26.20 |
| | Supernet Init | 389M | 61.00 | 30.00 | 24.02 | 51.54 | 38.23 | 26.21 | 26.35 | 18.84 | 65.34 | 24.57 | 51.38 | 33.11 | 60.73 | 39.33 | 23.66 |
| Pythia-12B | Random Init | 407M | 57.00 | 27.80 | 14.87 | 50.67 | 35.93 | 25.32 | 26.33 | 20.48 | 61.64 | 23.97 | 42.47 | 29.76 | 52.50 | 36.06 | 29.57 |
| | Supernet Init | 407M | 63.00 | 27.40 | 18.37 | 52.09 | 37.10 | 25.90 | 25.99 | 21.21 | 62.73 | 23.55 | 46.42 | 30.91 | 52.78 | 37.50 | 27.33 |
| Pythia-410M | Random Init | 405M | 62.00 | 29.60 | 19.54 | 50.67 | 36.49 | 25.72 | 25.54 | 20.15 | 64.14 | 24.57 | 47.35 | 32.70 | 61.01 | 38.42 | 25.29 |
| Pythia-6.9B | Random Init | 1.04B | 64.00 | 29.20 | 23.36 | 51.54 | 38.59 | 26.63 | 26.62 | 21.05 | 65.78 | 27.13 | 51.80 | 36.83 | 60.36 | 40.22 | 21.84 |
| | Supernet Init | 1.04B | 66.00 | 34.60 | 38.52 | 51.46 | 41.04 | 28.98 | 26.09 | 19.81 | 69.26 | 30.12 | 63.51 | 45.20 | 53.98 | 43.74 | 17.77 |
| Pythia-12B | Random Init | 1.04B | 63.00 | 28.40 | 23.33 | 50.51 | 37.92 | 27.05 | 26.14 | 19.57 | 66.76 | 26.36 | 53.74 | 36.65 | 51.77 | 39.32 | 21.21 |
| | Supernet Init | 1.04B | 64.00 | 31.20 | 27.56 | 51.77 | 38.48 | 27.36 | 26.19 | 19.82 | 66.54 | 26.45 | 53.96 | 36.42 | 61.65 | 40.88 | 20.77 |
| Pythia-1B | Random Init | 1.01B | 64.00 | 30.20 | 25.67 | 52.41 | 38.95 | 26.93 | 25.20 | 20.97 | 66.00 | 28.24 | 56.14 | 38.23 | 60.83 | 41.06 | 20.11 |
| Pythia-6.9B | Random Init | 2.91B | 61.00 | 30.60 | 26.49 | 52.10 | 39.00 | 27.60 | 26.39 | 19.82 | 67.74 | 28.33 | 57.83 | 41.12 | 59.05 | 41.31 | 13.75 |
| | Supernet Init | 2.91B | 66.00 | 34.60 | 50.16 | 56.91 | 41.71 | 30.54 | 26.45 | 20.80 | 72.69 | 33.87 | 67.09 | 53.40 | 62.05 | 47.41 | 10.99 |
| Pythia-12B | Random Init | 2.91B | 67.00 | 31.80 | 27.32 | 50.36 | 38.18 | 27.40 | 25.25 | 21.21 | 67.85 | 27.65 | 57.79 | 40.58 | 60.15 | 41.73 | 13.26 |
| | Supernet Init | 2.91B | 69.00 | 33.20 | 41.76 | 51.46 | 40.79 | 29.29 | 26.20 | 21.21 | 70.57 | 28.67 | 61.99 | 45.98 | 58.04 | 44.47 | 11.71 |
| Pythia-2.8B | Random Init | 2.78B | 68.00 | 30.4 | 24.51 | 53.03 | 39.50 | 27.18 | 25.74 | 20.47 | 67.68 | 25.34 | 46.67 | 39.11 | 59.54 | 40.55 | 14.54 |

Table 13: Evaluation of Pythia models across multiple benchmarks. Reported numbers are metrics as defined in Section 4.5 (%).

| Initialization | #Params | COPA | OpenBookQA | Lambada-OpenAI | Winogrande | Social IQA | MMLU-cont. | MMLU | CommonsenseQA | PIQA | ARC-challenge | ARC-easy | HellaSwag | BoolQ | Avg-acc | PPL-Nemotron-cc |
|---|---|---|---|---|---|---|---|---|---|---|---|---|---|---|---|---|
| from-supernet | 389M | 61.00 | 30.00 | 24.02 | 51.54 | 38.23 | 26.21 | 26.35 | 18.84 | 65.34 | 24.57 | 51.38 | 33.11 | 60.73 | 39.33 | 23.66 |
| from-supernet-distill | 389M | 66.00 | 30.60 | 23.95 | 49.57 | 37.41 | 26.31 | 25.62 | 19.57 | 65.56 | 25.34 | 49.66 | 33.92 | 54.31 | 39.06 | 18.59 |
| from-supernet | 1.04B | 66.00 | 34.60 | 38.52 | 51.46 | 41.04 | 28.98 | 26.09 | 19.81 | 69.26 | 30.12 | 63.51 | 45.20 | 53.98 | 43.74 | 17.77 |
| from-supernet-distill | 1.04B | 66.00 | 33.00 | 37.92 | 54.22 | 40.17 | 29.06 | 25.94 | 21.46 | 70.02 | 28.33 | 62.92 | 47.57 | 53.06 | 43.82 | 14.20 |

Table 14: Evaluation of sub-networks extracted from Pythia-6.9b for `bin-0` and `bin-1`. Reported numbers are metrics as defined in Section 4.5 (%). We compare training with the cross entropy loss (*from-supernet*) to training with knowledge distillation (*from-supernet-distill*) loss.

# E DETAILS ON IMPORTANCE SCORING

Importance scoring aims at defining scores for each transformer dimension, neuron or architecture parameter based on activation or weight magnitude. In our case, for a sub-network, the corresponding importance score serves as the proxy to sub-network quality or performance metrics like perplexity. The higher the importance score of a sub-network, the better its quality.

We adopt the dimension-wise importance scoring proposed by Muralidharan et al. (2024), which uses the activation of a component as proxy for its importance. Given a batch as input $X \in \mathbb{R}^{B \times T \times d_{model}}$ after applying the embedding layer $W^{emb}$ we compute the following scores for each component, where $B$ corresponds to the batch dimension and $T$ corresponds to the sequence length dimension, and $abs$ corresponds to the absolute value function:

- For a neuron $i \in \{1, ..., U\}$ in a FFN layer $l$, we compute its importance by: $F_{FFN_l}^{(i)} = 1/B \sum_B \left( 1/T \sum_T \mathbf{X} \mathbf{W}_1^l[:, i] \right)$ where $\mathbf{W}_1^l[:, i]$ corresponds to all weights of neuron $i$ in layer $l$.

- Similarly for each neuron $i \in \{1, ..., d_{model}\}$ in the embedding layer we compute $F_{emb}^{(i)} = 1/B \sum_B \left( 1/T \sum_T (\text{Norm}(\mathbf{X}[:, :, i])) \right)$. Specifically we perform mean absolute aggregation over output of every (Layer or RMS) Norm layer as

- For causal attention layers we compute the importance of head $h \in \{1, ..., H\}$ of heads as :

$$F_{\text{MHA}}^{(h)} = 1/B \sum_B \left( 1/T \sum_T \left\| \text{Attn}\left( \boldsymbol{Q}_h, \boldsymbol{K}_h, \boldsymbol{V}_h \right) \right\|_2 \right)$$

- For a block $l \in \{1, ..., L\}$ consisting of a MHA and a FFN layer with RMS or layer normalization in between, we compute the score: $F_{block}^{(l)} = 1 - 1/B \sum_B \left( 1/T \sum_T \left( \frac{\boldsymbol{X}_l^T \boldsymbol{X}_{l+1}}{\|\boldsymbol{X}_l\|_2 \|\boldsymbol{X}_{l+1}\|_2} \right) \right)$ where $\boldsymbol{X}_l$ is the input to block $l$ and $\boldsymbol{X}_{l+1}$ the output.

Given, the score for each unit (layer, head or neuron), and a subnetwork configuration (for example: $e = 64, d = 128, l = 2, h = 4$), we compute the importance score corresponding the sub-network, by simply aggregating the normalized importance scores corresponding to the selected neurons, layers or heads. Similarly, for the weight space we define neuron, layer and head level importance scores by simply focussing on the magnitude of weight or neurons corresponding to every transformer dimension (Han et al., 2015).

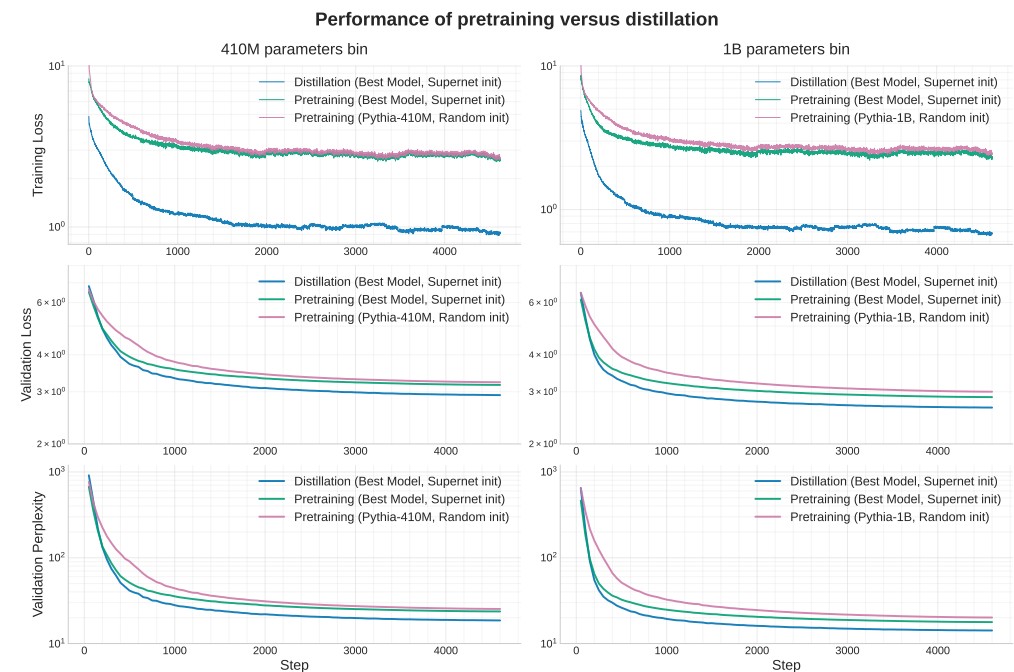

Figure 17: Training curves of the best models in bins 1 and 2, obtained through distillation and pretraining. In both cases, the models are initialized with weights from the supernet (Pythia-6.9B). For comparison, a Pythia model of similar size is trained from random initialization in both bins, serving as a baseline. All models are trained on 10 billion tokens.

---

**Algorithm 1** Bin-Constrained Evolutionary Search

---

1: **Input:** arch. space $\mathcal{S}$; bins $\{[L_b, U_b]\}_{b=1}^B$; population $N$; elites $k$; epochs $T$; random samples $r$; offspring $\lambda$; mutation prob. $m$; crossover prob. $c$
2: **Helpers:** $\mathrm{Params}(s)$ = param. count; $\mathsf{ppl}(s)$ = perplexity
3: CONSTRAIN$(x, [L, U])$: resample/repair until $\mathrm{Params}(x) \in [L, U]$
4: **for** $b = 1$ **to** $B$ **do**
5:    $\mathcal{S}_b \leftarrow \{s \in \mathcal{S} \mid L_b \leq \mathrm{Params}(s) \leq U_b\}$
6:    Init population $\mathcal{P}_b^{(0)} \sim \mathcal{S}_b$
7:    **for** $t = 0$ **to** $T - 1$ **do**
8:       Evaluate $\mathsf{ppl}(s)$ for $s \in \mathcal{P}_b^{(t)}$
9:       Select elites $\mathcal{E}_b^{(t)} = \arg\min^k \mathsf{ppl}(s)$
10:       Mutants $\mathcal{O}_{\mathrm{mut}} \leftarrow$ CONSTRAIN$(\mathrm{Mut}(s), [L_b, U_b])$, $s \sim \mathcal{E}_b^{(t)}$, size $\lambda$ (with prob. $m$)
11:       Crossovers $\mathcal{O}_{\mathrm{cross}} \leftarrow$ CONSTRAIN$(\mathrm{Cross}(s, s'), [L_b, U_b])$, $s, s' \sim \mathcal{E}_b^{(t)}$, size $\lambda$ (with prob. $c$)
12:       Randoms $\mathcal{R}_b^{(t)} \sim \mathcal{S}_b$, size $r$
13:       Next pop. $\mathcal{P}_b^{(t+1)} \leftarrow \arg\min^N \mathsf{ppl}(s)$ over $\mathcal{E}_b^{(t)} \cup \mathcal{O}_{\mathrm{mut}} \cup \mathcal{O}_{\mathrm{cross}} \cup \mathcal{R}_b^{(t)}$
14:    **end for**
15:    Best $s_b^\star \leftarrow \arg\min_{s \in \mathcal{P}_b^{(T)}} \mathsf{ppl}(s)$
16: **end for**
17: **Output:** $\{s_b^\star\}_{b=1}^B$

---

## F    WHITTLE APIS

An overview of the Whittle library is shown in Figure 10. In addition to the core functionalities of our framework described in Section 3, we provide an API to compute various importance metrics across different sub-network dimensions.

Listing 1: API for pretraining.

```python
def pretrain(
    model_name: str, # Name of the model to load. E.g., EleutherAI/pythia-410m
    model_config: Optional[Config] = None, # Model configuration, overrides model_name
    config_path : Optional[str] = None, # Path to yaml file with model configuration,
    overrides model_config
    out_dir: Path = Path("out/pretrain"), # Path to save checkpoints to
    precision: Literal["bf16-true", "bf16-mixed", "32-true", None] = None,
    resume: Union[bool, Literal["auto"], Path] = False, # If true, resumes from the
    latest available checkpoint
    data: Optional[DataModule] = None, # Dataset to use to train
    train: TrainArgs = TrainArgs( # Training hyperparameters
        save_interval=1000,
        log_interval=1,
        global_batch_size=512,
        micro_batch_size=4,
        max_tokens=int(3e12),  # 3 trillion
        max_norm=1.0,
        min_lr=4e-5,
        lr_warmup_steps=2000,
        tie_embeddings=False,
    ),
    eval: EvalArgs = EvalArgs(interval=1000, max_iters=100), # Evaluation hyper-
    parameters
    optimizer: Union[str, Dict] = "AdamW", # Optimizer and its configuration
    devices: Union[int, str] = "auto", # CUDA or CPU
    num_nodes: int = 1, # Number of nodes for distributed training
    tokenizer_dir: Optional[Path] = None, # Path to tokenizer (optional)
    logger_name: Literal["wandb", "tensorboard", "csv", "mlflow"] = "tensorboard", #
    Logger to use
    seed: int = 42, # Seed for reproducibility
    init_from: str = "random", # Path to the checkpoint to load, or "random" to
    randomly initialize
    use_flex: bool = False, # Set to True if the sub-network has layer-wise
    configuration
) -> LitGPT
```

Listing 2: API for supernetwork search.

```python
def search(
    supernet_name: str ="EleutherAI/pythia-12b",     # litgpt model to extract SLM from
    algorithm: str = "evolutionary_search",   # name of search algorithm
    num_bins: int = 4,                        # number of parameter bins
    param_upper_bounds: list,                 # list of upper bounds for bins
    param_lower_bounds: list,                 # list of lower bounds for bins
    number_of_epochs: int,                    # number of epochs for search
)-> list[dict]                                # returns list of subnets
```

**compute_importance_score().** The compute_importance_score() function (Listing 6, Appendix F) assigns an importance score to a given sub-network, where higher scores indicate higher estimated quality. Importance scores are computed independently for each architectural component (e.g., layers, heads, neurons), normalized across available choices using a softmax, and aggregated by summation. This computation is performed once at the beginning of the search procedure, after which evaluating the importance of candidate sub-networks becomes inexpensive compared to full metrics such as *perplexity*.

Below we present the details of our API design for setting subnetwork 4, pretrain 1, convert to litgpt 3, distillation 5, 2 search and importance metric computation 6.

Listing 3: API for converting a subnet into a LitGPT model.

```python
def convert_subnet_to_litgpt_model(
    supernet_name: str = "EleutherAI/pythia-12b",    # litgpt model to extract SLM from
    subnet_config: dict = {                          # subnet configuration
        sub_network_n_embd: 4,
        sub_network_intermediate_size: 16,
        sub_network_num_heads: 4,
        sub_network_n_layers: 2,
        sub_network_head_size: 4,
    }
) -> LitGPT                                           # returns LitGPT model
```

Listing 4: API for activating a sub-network

```python
def set_sub_network(
    sub_network_n_embd: int = 4, # Embedding dim
    sub_network_intermediate_size: int | list[int] = 42  # MLP size
    sub_network_num_heads: int | list[int] = 4, # No. of Attention heads
    sub_network_n_layers: int = 4, # No. of Layers
    sub_network_query_groups: int | list[int] = 2, # No. of Query groups
    sub_network_head_size: int | list[int] = 4,  # Head size
    sampled_intermediate_indices: list[int] | list[list] = [4,8], # Sampled MLP neurons
    sampled_head_indices: list[int] | list[list] = [2,3], # Sampled heads
    sampled_query_group_indices: list[int] | list[list] = [0,1], # Sampled query groups
    sampled_head_size_indices: list[int] | list[list] = [2,3,8,12], # Sampled head sizes
    sampled_layer_indices: list[int] = [2,3,5,6], # Sampled layers
    sampled_embd_indices: list[int] = [0,1,2,3], # Sampled embedding neurons
)
```

# G    STUDYING DISTILLATION HYPERPARAMETERS

We define our distillation loss function below.

$$\mathcal{L} = \alpha \, \mathcal{L}_{\text{CE}}(\mathbf{y}, \mathbf{s}) + \beta \sum_{i \in \mathcal{K}} \text{KL}\left( \text{softmax}\left( \tfrac{z_t^{(i)}}{T} \right) \,\|\, \text{softmax}\left( \tfrac{z_s^{(i)}}{T} \right) \right)., \tag{5}$$

This loss has four tunable hyperparameters:

1. $\alpha$: the weight of the cross-entropy loss, which minimizes the entropy with respect to the ground-truth logits.

2. $\beta$: the weight of the KL-divergence term between the teacher and student logits.

3. $T$: the temperature parameter, which controls the smoothing of the teacher and student logit distributions.

4. $\mathcal{K}$: the number of top logits (Top-$\mathcal{K}$) used when computing the KL-divergence. A smaller $\mathcal{K}$ corresponds to a simpler distribution, while a larger $\mathcal{K}$ yields a more informative distribution. The maximum $\mathcal{K}$ equals the full number of teacher logits.

We now perform a grid-sweep over different choices of $\alpha$, $\beta$, $T$ and $\mathcal{K}$. We define the set of choices for $\alpha$ as $[0.2, 0.8, 0]$, the corresponding choices for $\beta$, which corresponds to $1 - \alpha$ as $[0.8, 0.2, 1]$, the choices for temperature $T$ as $[0.8, 0.9]$ and the choices for $\mathcal{K}$ as $[1024, 2048, num\_teacher\_logits]$. Figures 18-20 present the perplexity curves aggregated for different hyperparameter values. In general, we observe that using only the distillation loss, i.e., setting $\alpha = 0$, is not recommended. Furthermore, a higher temperature and using the full logit distribution (Top-K = 0) perform best on average. In Table 15, we also present the importance of each of the hyperparameter choices and find that the most important one is the value of $\alpha$, followed by $\mathcal{K}$ and finally the temperature $T$.

Listing 5: API for distilling a sub-network from a checkpoint.

```python
def distill(
    teacher_checkpoint_dir: Path, # Path to teacher model checkpoint directory
    student_dir: Path, # Path to initialize student model directory with sub-network
    configuration and (optional) model weights
    data: DataModule | None = None, # Dataset for distillation
    out_dir: Path = Path("out/distill"), # Path to save distilled checkpoints
    precision: Literal["bf16-true", "bf16-mixed", "32-true", None] = None, # Precision
    for training
    train: TrainArgs = TrainArgs( # Training hyperparameters
        save_interval=1000,
        log_interval=1,
        global_batch_size=512,
        micro_batch_size=4,
        max_tokens=int(5e8),
        max_norm=1.0,
        min_lr=4e-5,
        lr_warmup_steps=2000,
        tie_embeddings=False,
    ),
    distill: DistillArgs = DistillArgs( # Distillation-specific hyperparameters
        method="logits", # Distillation method (e.g., logits, hidden states)
        temperature=10,  # Softening factor for teacher logits
        alpha=0.3,       # Weight for student loss
        beta=0.7,        # Weight for distillation loss
        loss="forward_kld", # Loss function for distillation
        weight_scheme="other", # Weighting scheme for combining losses
    ),
    eval: EvalArgs = EvalArgs(interval=50, max_iters=100, initial_validation=True), #
    Evaluation config
    optimizer: str | dict = "AdamW", # Optimizer and configuration
    devices: int | str = "auto", # CUDA or CPU
    num_nodes: int = 1, # Number of nodes for distributed distillation
    tokenizer_dir: Path | None = None, # Path to tokenizer (optional)
    logger_name: Literal["wandb", "tensorboard", "csv"] = "csv", # Logger backend
    seed: int = 42, # Seed for reproducibility
    random_init_student: bool = False, # If True, randomly initialize student instead
    of loading
) -> LitGPT # Returns a distilled LitGPT student model
```

Listing 6: API for computing importance scores of subnet components.

```python
def compute_importance_score(
    supernet_name: str = "EleutherAI/pythia-12b",   # base supernetwork
    subnet_config: dict = {                          # subnet configuration
        sub_network_n_embd: 4,
        sub_network_intermediate_size: 16,
        sub_network_num_heads: 4,
        sub_network_n_layers: 2,
        sub_network_head_size: 4,
    },
    layer_importance_type: str  = "block importance", # method for layer scoring
    head_importance_type: str   = "minitron",         # method for head scoring
    neuron_importance_type: str = "minitron",         # method for neuron scoring
) -> int:   # returns importance score for a sampled sub-network
```

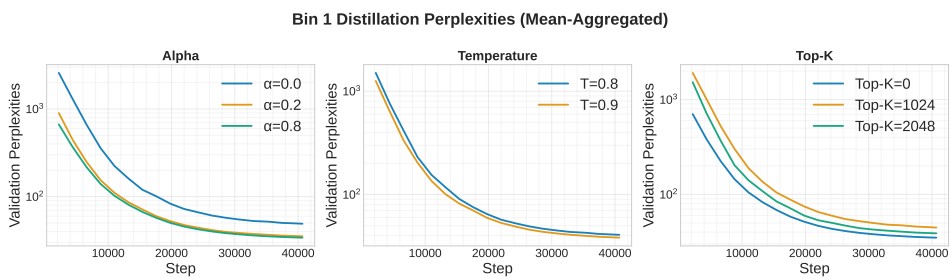

Figure 18: Bin 1 - distillation hyperparameter importance

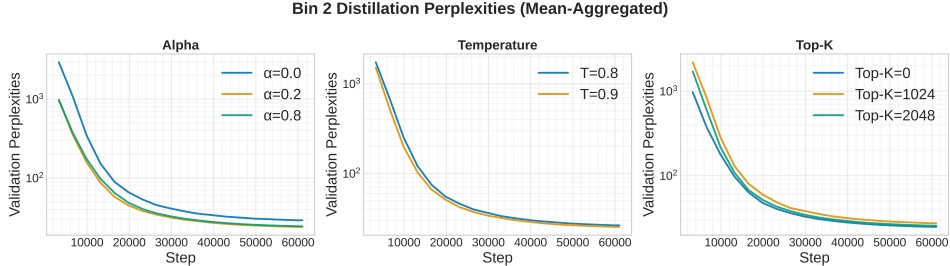

Figure 19: Bin 2 - distillation hyperparameter importance

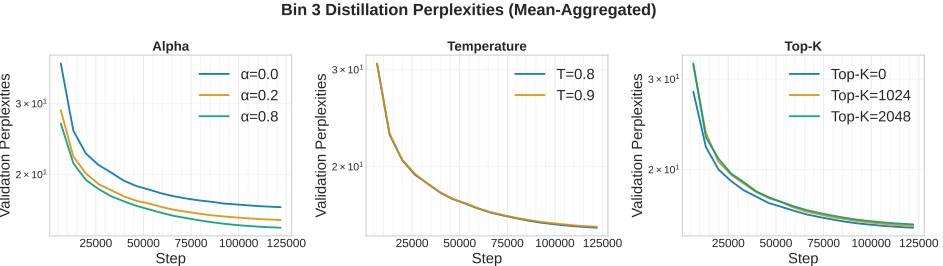

Figure 20: Bin 3 - distillation hyperparameter importance

| hyperparameter | importance (mean) | importance (std) |
|---|---|---|
| $\alpha$ | 0.986442 | 0.011966 |
| $\mathcal{K}$ | 0.005968 | 0.000469 |
| $T$ | 0.005000 | 0.000551 |

Table 15: Hyperparameter Importance

# H  SEARCH SPACE SIZES

The sizes of the search spaces for evolutionary search, especially for fine-grained (both uniform and layer-wise) can grow exponentially. In Table 16, we show the maximum number of configurations per search space for both of our base Pythia models (6.9B and 12B). For equations and information on how the number of search space configurations is calculated, refer to Section 2.1, and Table 1 for the configurations.

Table 16: Search Space Sizes ($N$) for Pythia-6.9B and Pythia-12B Architectures

| Search Space | Pythia-6.9B ($N$) | Pythia-12B ($N$) |
|---|---|---|
| Coarse Uniform | $4.33 \times 10^{12}$ | $9.51 \times 10^{12}$ |
| Coarse Layer-wise | $1.65 \times 10^{244}$ | $2.33 \times 10^{281}$ |
| Fine-grained Uniform | $10^{14284}$ | $10^{17841}$ |
| Fine-grained Layer-wise | $1.58 \times 10^{159984}$ | $5.31 \times 10^{224611}$ |

# I  REPRODUCIBILITY STATEMENTS

We have taken extensive measures to ensure that all results in this paper can be replicated and verified by the community.

- **Code and Repository:** We release all our code and scripts to reproduce our experiments at `https://anonymous.4open.science/r/whittle-iclr-71CD/`.
- **Datasets and Pretrained Models:** We evaluate on available benchmarks from lm-eval-harness (`https://github.com/EleutherAI/lm-evaluation-harness`) and use the publicly available Nemotron-CC dataset `https://research.nvidia.com/labs/adlr/Nemotron-CC/` for training. Furthermore we use the Pythia-model suite, which is open-source `https://github.com/EleutherAI/pythia`.
- **Compute Resources:** All our search experiments were run on on L40 GPU per parameter bin and base model. All our pretraining runs for `bin-0` and `bin-1` were run on 8 L40 GPUs and `bin-2` was run on 4 H200 GPUs. All our distillation experiments were run on 4 H200 GPUs. We use cuda version 11.8.
- **Evaluation and Artifacts:** Upon acceptance of the paper we will publicly release model checkpoints for all our experiments.

# J  SCALING BEHAVIOR OF SUBNETWORK EXTRACTION UNDER LARGER BUDGETS

To assess how the cost savings from subnetwork extraction scale with substantially larger pretraining budgets, we trained the best model from Bin 2 and Bin 3 for 100 billion tokens and compared it against a Pythia-1B and Pythia-2.8B model, respectively, trained for the same number of tokens from random initialization. We summarize our key findings below.

- **Sustained FLOP Savings at the 100B-Token Scale**. Our extracted subnetwork for Bin-3 achieves the same validation performance while requiring $1.26\times$ fewer FLOPs (a reduction

| Base Model | Initialization | #Params | COPA | OpenBookQA | Lambada-OpenAI | Winogrande | Social IQA | MMLU-cont. | MMLU | CommonsenseQA | PIQA | ARC-challenge | ARC-easy | HellaSwag | BoolQ | Avg-acc |
|---|---|---|---|---|---|---|---|---|---|---|---|---|---|---|---|---|
| Pythia-6.9B | Supernet Init | 1.04B | 74.00 | 38.40 | 45.604 | 56.98 | 41.81 | 32.54 | 27.04 | 20.88 | 75.03 | 38.57 | 71.76 | 60.05 | 61.19 | 49.53 |
| Pythia-1B | Random Init | 1.01B | 71.00 | 35.40 | 36.13 | 53.35 | 41.91 | 26.93 | 25.20 | 19.74 | 73.50 | 35.92 | 69.36 | 55.69 | 52.75 | 45.91 |
| Pythia-6.9B | Supernet Init | 2.91B | 76.00 | 37.40 | 53.23 | 58.01 | 42.02 | 34.45 | 38.04 | 43.41 | 77.69 | 41.38 | 73.57 | 64.01 | 63.49 | 54.05 |
| Pythia-2.8B | Random Init | 2.78B | 71.00 | 40.40 | 43.39 | 58.64 | 42.02 | 27.18 | 25.74 | 21.37 | 76.50 | 42.75 | 74.24 | 64.85 | 60.58 | 49.23 |

Table 17: Evaluation of Pythia models trained for 100B tokens across multiple benchmarks. Reported numbers are metrics as defined in Section 4.5 (%).

of approximately 21%). Although this reduction is smaller than the $5.16\times$ savings observed at the 10B-token scale, it nevertheless demonstrates that subnetwork extraction continues to provide meaningful computational benefits even when the training budget is increased by an order of magnitude. This indicates that the method is not confined to low-budget regimes and remains competitive at significantly larger compute settings.

- **Improved Final Validation Perplexity**. In addition to being more compute-efficient, the extracted model also attains a lower final validation perplexity. The baseline Pythia-2.8B reaches a perplexity of 11.397, while our model achieves 11.204.

- **Strong Downstream Performance Advantages.** The extracted model outperforms the Pythia models trained from scratch across downstream evaluations. In particular, on MMLU-cont, our model achieves gains of up to 12% over the strongest Pythia baseline (see Table 17).

In Figures 21-23, we present the trajectories for validation perplexity, validation loss and train loss, respectively, for Bin 2 and Bin 3 architectures and the Pythia-based models trained from scratch for 100B tokens.

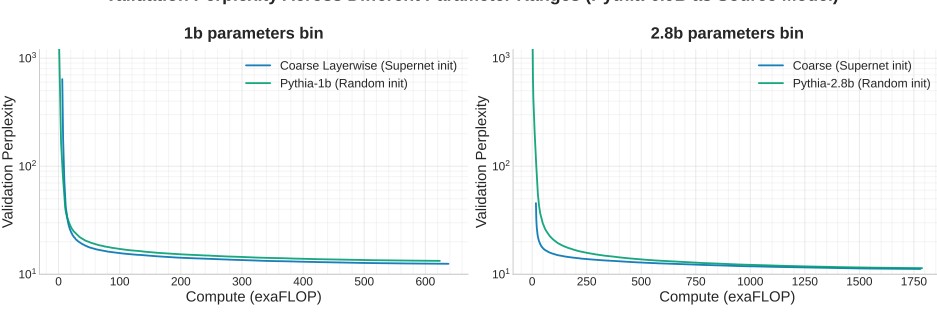

Figure 21: Bin 2 and Bin 3 validation perplexity for 100B token budget

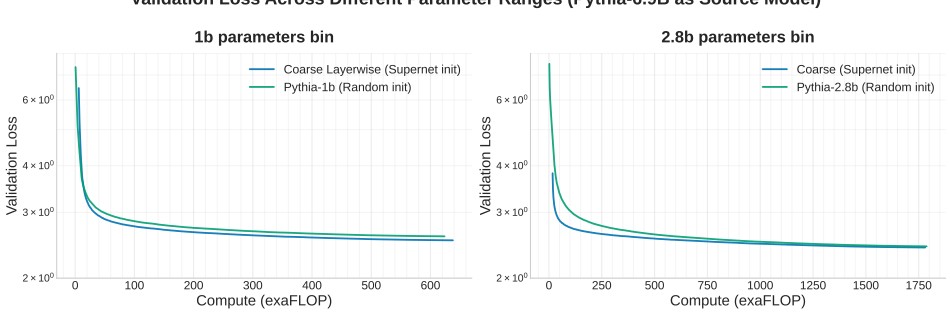

Figure 22: Bin 2 and Bin 3 validation loss for 100B token budget

Figure 23: Bin 2 and Bin 3 train loss for 100B token budget

