# OpenReview forum: "Where to Begin: Efficient Pretraining via Sub-network Selection and Distillation"
_ICLR.cc/2026/Conference — Submitted to ICLR 2026_

### Official Review · Reviewer_5QXR · 2025-10-21

**Soundness:** 3
**Presentation:** 3
**Contribution:** 2
**Rating:** 4
**Confidence:** 4

**Summary:**

This paper addresses the high cost of pretraining Small Language Models by proposing a framework that combines three synergistic techniques: 1) warm-starting the SLM by extracting an initial sub-network from a large, pretrained teacher LLM; 2) using evolutionary search to find the optimal sub-network architecture within a parameter budget; and 3) applying knowledge distillation from the teacher during pretraining. The authors demonstrate that this framework is highly effective, with their best model achieving the same validation perplexity as 9.2× fewer pretraining tokens.

**Strengths:**

1. Significant Practical Impact: The 9.2× token-efficiency saving is a major achievement, making high-performance SLM development accessible with limited compute.
2. Contribution to Reproducibility: By releasing the whittle library, the authors provide a clear, generalizable pipeline, addressing the closed-source limitations of prior art and contributing significantly to the community.

**Weaknesses:**

1. Incomplete Discussion of Related Work: The paper is actually very similar to methods that prune and then continue pretraining to recover performance, which is an active area of research. However, the authors appear to have not discussed any related papers, and the performance improvements they achieve are also similar. The authors should compare their work to Sheared LLaMA, DRPruning and other recent methods to clarify their unique contribution. Therefore, I believe this paper's contribution to the community is relatively limited.

[1] Sheared LLaMA: Accelerating Language Model Pre-training via Structured Pruning. ICLR 2024.

[2] DRPruning: Efficient Large Language Model Pruning through Distributionally Robust Optimization. ACL 2025.

2. Unaccounted Cost of Evolutionary Search: The paper's core claim is "efficiency." However, the evolutionary search itself consumes significant compute. The 9.2× token saving figure does not account for this search cost. If not, the true end-to-end efficiency gain is overstated.

**Questions:**

See Weaknesses.

---

> ### Author Response · Authors · 2025-11-23
> **Official Response to Reviewer 5QXR**
>
> Thank you for your thoughtful and detailed evaluation of our work. We appreciate the recognition of the strengths of our approach, including the ability to train SLMs under limited compute budgets and the usefulness of our whittle library, which simplifies extracting SLMs from several open-source LLMs. We respond to each of your questions below:
>
> > Incomplete discussion of Related Work
>
> We thank the reviewer for pointing out other relevant works such as Sheared LLaMA [1] and DRPruning [2]. We compare and contrast each of these with our work below:
>
> **Whittle vs. Sheared LLaMA.**
>
> - **Obtaining an SLM**: Sheared LLaMA frames pruning as a constrained optimization problem that jointly updates weights and pruning masks to match a target architecture. As noted in Section 2.1, this requires repeated pretraining-style updates and is therefore computationally expensive. In contrast, given upper and lower bounds on parameter count, our evolutionary search directly evaluates candidate architectures using only forward passes to estimate perplexity. This avoids weight/mask updates and leads to substantially lower compute and memory cost than the structured-pruning procedure used in Sheared LLaMA.
> - **Pretraining an SLM**: Sheared LLaMA further employs dynamic batch loading, where batches are selected based on loss-reduction rate, making it difficult to disentangle improvements from pruning vs. this training strategy. Whittle instead uses standard next-token-prediction pretraining, matching the training procedure of models such as Pythia and LLaMA. Because the pruning gains in Sheared LLaMA are confounded with its dynamic batch-loading procedure, we do not view it as a clean baseline for our setting.
>
> **Whittle vs. DRPruning.**
>
> DRPruning, like Sheared LLaMA, performs structured pruning by learning pruning masks at multiple granularities. It introduces a distributionally robust pruning scheme that dynamically adjusts data ratios during training to improve robustness under distribution shift. This approach is computationally expensive because each mask requires optimization under the full pretraining objective. In addition, the dynamic data selection makes it difficult to disentangle the benefits from pruning versus those from data reweighting.
>
> In contrast, Whittle is a simple, modular library that identifies the best SLM initialization within a target parameter range using only forward-pass evaluations. It outputs architectures that can be directly plugged into any standard LLM pretraining pipeline, without additional training complexity or confounding factors.
> We will add this extended discussion comparing whittle to pruning methods to the updated version of the paper.
>
> > Unaccounted Cost of Evolutionary Search
>
> Thank you for highlighting this important point. We agree that, for a fair comparison, the computational cost of the evolutionary search should also be taken into account. Below, we report the estimated search cost (in floating point operations) for each bin:
>
> | Bin    | Cost        |
> |--------|-------------|
> | 1 | 2.3 exaFLOP  |
> | 2 | 5.4 exaFLOP  |
> | 3 | 15.4 exaFLOP |
>
> For comparison, the cost of pretraining the best model found in each bin for 10 billion tokens is:
>
> | Bin    | Cost        |
> |--------|-------------|
> | 1 | 20.9 exaFLOP  |
> | 2 | 63.3 exaFLOP  |
> | 3 | 176.6 exaFLOP |
>
> As you can see, the search cost is only a fraction of the total pretraining cost.
>
> After adding the search cost to the pretraining cost, the revised numbers are follows:
> | Bin    | FLOP savings factor       |
> |--------|-------------|
> | 1 | 1.71x |
> | 2 | 1.75x |
> | 3 | 5.16x |
>
> The revised figure, which adds the search cost to the pretraining budget, can be found [here](https://figshare.com/s/2813da9b9bc7cbf52520?file=59778620).
>
> We will update the paper with these new figures and estimates.
>
> We would like to thank you again for the very detailed review. We hope that we were able to address your concerns and that you would consider increasing your score. We are happy to engage in further discussions with you and address any further questions you may have during the rebuttal period.

---

### Official Review · Reviewer_KKny · 2025-10-25

**Soundness:** 3
**Presentation:** 3
**Contribution:** 3
**Rating:** 8
**Confidence:** 4

**Summary:**

The authors introduce a novel method to train efficiently SLM's leveraging a LLM for initialization and distillation.
They provide a method to search the subnetwork space of a LLM corresponding to a space of possible SLM architectures constrained by a smaller parameter budget. They use an evolutionary algorithm to search this space and retrieve the most promising subnetwork, which is then pre-trained with a distillation objective using the same LLM as teacher. They accelerate the SLM pre-training by up-to 9.2X compared with a SLM with the same architecture but randomly initialized. They open-source a library to reproduce and extend the work.

**Strengths:**

* The idea to use an evolutionary algorithm to extract a subnetwork as initialization is novel and convincing
* They evaluate their strategy thoroughly and obtain impressive speedups to train SLM
* They open source a library to reproduce and extend their work

**Weaknesses:**

Not seeing any significant weakness

**Questions:**

* The method seems generic. What are the constrained for a neural network architecture so that the method can be applied to any type of network (e.g. ResNets and image classification)?

*  If the comparison between training from scratch a SLM with random initialization and pre-training it from the retrieved LLM subnetwork also include the LLM subnetwork search budget, will the 9.2x speedup persist? How does this change the comparison?

---

> ### Author Response · Authors · 2025-11-23
> **Response to Reviewer KKny**
>
> Thank you for carefully reading our paper, your feedback and the positive score. We are  encouraged to see that you find our idea novel and convincing, our evaluation strategy thorough and the speedups obtained impressive. Furthermore, we appreciate your recognition of our released open-source, reproducible framework whittle, which is the primary contribution of our work. We respond to each of your questions and concerns below:
>
>
> > Generality of the method and potential application to image classification
>
> Thank you for the question. Our method is general and can, in principle, be applied to a wide range of architectures. For example, in the case of ResNets, one could search over depth, number of channels, kernel sizes, and related architectural choices. However, our library and experiments focus on LLMs, where the high pretraining cost makes selecting an appropriate subnetwork for a given parameter budget particularly valuable. Extending whittle to the vision domain is beyond the scope of the current work, but we view it as an exciting direction for future research.
>
> > Accounting for cost of subnetwork search
>
> Thank you for highlighting this important point. We agree that, for a fair comparison, the computational cost of the evolutionary search should also be taken into account. Below, we report the estimated search cost (in floating point operations) for each bin:
>
> | Bin    | Cost        |
> |--------|-------------|
> | 1 | 2.3 exaFLOP  |
> | 2 | 5.4 exaFLOP  |
> | 3 | 15.4 exaFLOP |
>
> For comparison, the cost of pretraining the best model found in each bin for 10 billion tokens is:
>
> | Bin    | Cost        |
> |--------|-------------|
> | 1 | 20.9 exaFLOP  |
> | 2 | 63.3 exaFLOP  |
> | 3 | 176.6 exaFLOP |
>
> As you can see, the search cost is only a fraction of the total pretraining cost.
>
> After adding the search cost to the pretraining cost, the revised numbers are follows:
> | Bin    | FLOP savings factor       |
> |--------|-------------|
> | 1 | 1.71x |
> | 2 | 1.75x |
> | 3 | 5.16x |
>
> The revised figure, which adds the search cost to the pretraining budget, can be found [here](https://figshare.com/s/2813da9b9bc7cbf52520?file=59778620).
>
> We will update the paper with these new figures and estimates.
>
> Thank you again for the review. We hope our clarifications adequately address your concerns . We would be glad to engage further and answer any additional questions throughout the rebuttal phase.

---

> > ### Comment · Reviewer_KKny · 2025-11-26
> >
> > Thank you for your satisfying answers. As a result, I am maintaining my rating as a good paper to be accepted.

---

### Official Review · Reviewer_HjZw · 2025-10-30

**Soundness:** 2
**Presentation:** 2
**Contribution:** 2
**Rating:** 4
**Confidence:** 4

**Summary:**

This paper introduces a framework to make the pretraining of Small Language Models (SLMs) more efficient. The core idea is to first select a smaller sub-network from a large, pretrained teacher model and use its weights as a warm start. This sub-network is then further trained, primarily using knowledge distillation from the teacher model. The authors employ an evolutionary search algorithm to find optimal sub-network architectures within different parameter budgets. They also release an open-source library called `whittle` to facilitate this process. The results show that this approach can significantly reduce the number of pretraining tokens required to reach a target performance level compared to training a standard model of a similar size from scratch.

**Strengths:**

1. The paper provides some useful insights into the process of knowledge distillation for SLMs.

2. The paper explores four types of search spaces, from coarse to fine-grained and from uniform to layer-wise configurations.

**Weaknesses:**

1. The main contribution of the paper feels incremental. The general strategy of first pruning a large model and then using distillation to train the smaller model has been explored in previous works, such as Sheared LLaMA [1] and Minitron [2].

2. The proposed sub-network selection method, particularly in the fine-grained search spaces, seems to lack sophistication. The paper states that for selecting components like attention heads or neurons within a layer, it samples indices from the teacher's components. This appears to be a random selection process that does not consider weight importance, activation patterns, or any other semantic information, which are common considerations in modern structured pruning methods.

3. The performance comparison is not convincing enough. To better demonstrate the effectiveness of the proposed method, it should be benchmarked against strong, relevant baselines like Sheared LLaMA [1], which also focuses on efficient pre-training through structured pruning. Without this comparison, it is difficult to assess the true advantage of this framework over the state-of-the-art.

**References**

[1] Sheared LLaMA: Accelerating Language Model Pre-training via Structured Pruning.

[2] LLM Pruning and Distillation in Practice: The Minitron Approach.

**Questions:**

1.  The experiments are conducted on the Pythia model family. Would this method work as effectively on more powerful base models, such as the Qwen family? It is possible that stronger, more optimized models are inherently harder to compress and might not yield the same efficiency gains.

2.  Could you provide more detailed statistics on the overhead and efficiency of the search phase? The paper mentions a budget of 16 hours per parameter bin. It would be helpful to see more granular data to better understand the costs of this initial search step.

3.  How is the crossover operation performed in the fine-grained search space?

---

> ### Author Response · Authors · 2025-11-23
> **Response to Reviewer HjZw**
>
> Thank you for a thorough and thoughtful review of our paper. We appreciate that you highlight different positive aspects of our work, including insights about distillation for SLMs and exploration of search spaces at different levels of granularities. We respond to each of your questions and concerns below:
>
> > The main contribution of the paper feels incremental..
>
> Thank you for this feedback. While prior work has indeed explored pruning larger models followed by distillation, we would like to clarify that this is not the primary contribution of our paper. Our main contribution is **whittle**—a modular library that enables structural pruning (and optional distillation) for **LitGPT** models. **LitGPT** is a highly active and widely used LLM ecosystem (12.9k GitHub stars) that continuously updates its supported **HuggingFace** models. Because **whittle** is built as a wrapper around **LitGPT**, it automatically inherits support for all models in the **LitGPT** ecosystem while adding flexible subnetwork/SLM search capabilities.
>
> Notably, **whittle** allows pruning at the most granular level (individual neurons) and outputs SLMs that can be directly consumed by standard **LitGPT**-style pretraining pipelines. Thus, our contribution is a practical and extensible library for effective SLM extraction and pretraining, rather than introducing a new pruning–distillation method.
>
> > The proposed sub-network selection method seems to lack sophistication
>
> We agree with your observation. Our implementation of evolutionary search is intentionally simple: mutations and crossovers are applied to a randomly sampled population without incorporating any domain-specific heuristics. More informed mutation strategies could indeed yield better architectures—or achieve similar performance with fewer search iterations.
>
> However, we emphasize that the evolutionary search procedure itself is not the primary contribution of our work. Evolutionary search is merely one instantiation of a generic search algorithm and can be replaced by alternatives such as random search or Bayesian optimization.
>
> Our central contribution is whittle, a flexible library that enables users to extract effective SLMs from pretrained LLMs. The framework is designed so that users can easily plug in more sophisticated search strategies, which may improve results. To demonstrate the utility of the library, we intentionally use a simple selection mechanism, as the reviewer notes; the fact that this already yields compelling results highlights the value of a framework that supports such search capabilities.
>
> Additionally, in Section 5 (Ablations), we explore alternative semantic signals, such as Minitron Importance and Magnitude Importance to guide the search, and find that perplexity remains the most effective metric. We acknowledge that this differs from incorporating semantic information directly into mutation and crossover operations, which we leave for future work.
>
> > The performance comparison is not convincing enough and comparison with Sheared-Llama [1]
>
> Thank you for mentioning the related work. Sheared-Llama [1] differs significantly from whittle and we delineate the primary differences below:
>
> - **Obtaining an SLM**: Sheared LLaMA frames pruning as a constrained optimization problem that jointly updates weights and pruning masks to match a target architecture. As noted in Section 2.1, this requires repeated pretraining-style updates (for every target architecture structure) and is therefore computationally expensive. In contrast, given upper and lower bounds on parameter count, our evolutionary search directly evaluates candidate architectures using only forward passes to estimate perplexity. This avoids weight/mask updates and leads to substantially lower compute and memory cost than the structured-pruning procedure used in Sheared LLaMA.
> - **Pretraining an SLM**: Sheared LLaMA further employs dynamic batch loading, where batches are selected based on loss-reduction rate, making it difficult to disentangle improvements from pruning vs. this training strategy. Whittle instead uses standard next-token-prediction pretraining, matching the standard training procedure of models such as Pythia and LLaMA.
>
> Since the pruning gains in Sheared LLaMA are confounded with its dynamic batch-loading procedure, we do not view it as a clean baseline for our setting.

---

> ### Author Response · Authors · 2025-11-23
> **Continuation of response to Reviewer HjZw**
>
> > Generalizability to architectures like Qwen and LLaMa
>
> Thank you for this important question. Methodologically, since *whittle* is implemented as a wrapper around LitGPT, it immediately supports all HuggingFace models integrated into LitGPT (an exhaustive list is available [here](https://github.com/Lightning-AI/litgpt/tree/v0.5.11?tab=readme-ov-file#choose-from-20-llms)), including model families such as LLaMA and Qwen. As the reviewer notes, this framework opens the door to several interesting research directions—for example, studying how LLM architecture type, data quality, or model optimization level affect compressibility. However, because our primary contribution in this work is the *whittle* library itself, these broader research questions lie beyond the current scope and remain promising avenues for future work.
>
> Following the reviewer’s suggestion, we are now pretraining SLMs extracted by *whittle* from the LLaMA-3.1-8B models, and we will update the reviewer as soon as these results become available.
>
> >  Overhead and efficiency of the search procedure
>
> Thank you for this question. In each bin, for every search space, we sampled and evaluated 5050 subnetworks as part of the evolutionary search. The perplexity of these models on 1000 sequences of length 512 were computed. If we approximate the FLOP for Pythia-410m, Pythia-1b, and Pythia-2.8b as the average FLOP per model in bin 1, 2, and 3, then the total computational cost are as follows:
> | Bin    | Cost        |
> |--------|-------------|
> | 1 | 2.3 exaFLOP  |
> | 2 | 5.4 exaFLOP  |
> | 3 | 15.4 exaFLOP |
>
> For comparison, the cost of pretraining the best model found in each bin for 10 billion tokens is:
>
> | Bin    | Cost        |
> |--------|-------------|
> | 1 | 20.9 exaFLOP  |
> | 2 | 63.3 exaFLOP  |
> | 3 | 176.6 exaFLOP |
> So, the search phase costs only a fraction of the pretraining cost.
>
> Based on the feedback from the other reviewers, we have now accounted for the cost of the evolutionary search in computing the cost savings during the pretraining of the models. The revised estimates are now as follows:
>
> | Bin    | FLOP savings factor       |
> |--------|-------------|
> | 1 | 1.71x |
> | 2 | 1.75x |
> | 3 | 5.16x |
>
> The revised figure, which adds the search cost to the pretraining budget, can be found [here](https://figshare.com/s/2813da9b9bc7cbf52520?file=59778620).
>
> We will update the paper with these new figures and estimates.
>
> > Crossover operation performed in the fine-grained search space?
>
> Thank you for this question. We retain the crossover operation within the fine-grained search space. Specifically, we first select two parent architectures that have the same number of layers. Given such parents, \(p_1\) and \(p_2\), we perform a coin flip to determine, for each architectural property (e.g., the intermediate dimension of each layer), whether the child inherits this property from \(p_1\) or \(p_2\).
> In the fine-grained search space, we additionally copy the corresponding sampled neurons for each dimension of every layer from the parent selected by the coin flip. Thus, for each architectural dimension at each layer, the child inherits properties from either \(p_1\) or \(p_2\), determined independently by the stochastic crossover mechanism.
>
> We would like to thank you again for the detailed review. We hope that we were able to address your concerns and that you would consider increasing your score. We are happy to engage in further discussions with you and address any further questions you may have during the rebuttal period.

---

### Official Review · Reviewer_4Dsu · 2025-11-03

**Soundness:** 3
**Presentation:** 3
**Contribution:** 3
**Rating:** 6
**Confidence:** 4

**Summary:**

The paper presents a methodology to initialize language models from larger pretrained models using evolutionary search and subsequently train them with distillation to achieve high-quality small language models. In particular, the authors present the search space they consider and then an evolutionary search algorithm to identify good candidates in that search space. They show in their experiments that the networks found using their search algorithm perform significantly better than random initialization or random points in that search space.

**Strengths:**

- The paper is very well written with clearly defined goal and methodology as well as a comprehensive appendix.
- The evolutionary search algorithm is clearly presented and intuitive.
- The results clearly show that starting from a model found using the evolutionary search is a significantly better choice compared to random initialization and a random point in the search space.

**Weaknesses:**

The main weakness of the paper is in its evaluation section. In more detail:

1. When comparing initialization performance of different methods (superset, random init, etc) none of the methods are actually FLOP normalized. Namely, the evolutionary search required some FLOPS to identify the best model. However, random initializations didn't so they should be allowed more optimization to account for that discrepancy. More importantly, the evolutionary search should be compared to random search ie sample different points from the search space and keep the best.
2. It is slightly worrying that the bigger the model the coarser the search space needs to be. The more fine-grained search spaces are supersets of the coarse ones so they should perform at least as well. The fact that they don't points towards the search not being effective which increases the importance of point 1.

**Questions:**

I have laid out my concerns in the weaknesses section.

---

> ### Author Response · Authors · 2025-11-23
> **Response to Reviewer 4Dsu**
>
> Thank you for your thoughtful evaluation of our work and the positive score. We appreciate that you find our paper well written, the methodology and the goal well defined and the evolutionary search intuitively presented. We respond to each of your questions below:
>
> >  FLOP normalization
>
> Thank you for highlighting this important point. We agree that, for a fair comparison, the computational cost of the evolutionary search should also be taken into account. Below, we report the estimated search cost (in floating point operations) for each bin:
>
> | Bin    | Cost        |
> |--------|-------------|
> | 1 | 2.3 exaFLOP  |
> | 2 | 5.4 exaFLOP  |
> | 3 | 15.4 exaFLOP |
>
> For comparison, the cost of pretraining the best model found in each bin for 10 billion tokens is:
>
> | Bin    | Cost        |
> |--------|-------------|
> | 1 | 20.9 exaFLOP  |
> | 2 | 63.3 exaFLOP  |
> | 3 | 176.6 exaFLOP |
>
> As you can see, the search cost is only a fraction of the total pretraining cost.
>
> After adding the search cost to the pretraining cost, the revised numbers are follows:
> | Bin    | FLOP savings factor       |
> |--------|-------------|
> | 1 | 1.71x |
> | 2 | 1.75x |
> | 3 | 5.16x |
>
> The revised figure, which adds the search cost to the pretraining budget, can be found [here](https://figshare.com/s/2813da9b9bc7cbf52520?file=59778620).
>
> We will update the paper with the new figures and estimates.
>
>
> > Comparison of Evolutionary Search (ES) with Random Search (RS)
>
> Thank you for the suggestion to compare evolutionary search (ES) with random search (RS). Whittle already supports multiple search algorithms (e.g., those available in Syne-Tune) and can be easily extended to incorporate additional methods. Following the reviewer’s recommendation, we are currently pretraining architectures obtained via RS and comparing them with those discovered by ES on a 2B-token budget. We will update the results here as soon as they become available.
>
> > Finegrained v/s Coarse grained search spaces
>
> Thank you for pointing this out. To provide clearer context, we present a table reporting the search-space sizes for both Pythia-6.9B and Pythia-12B below. While the fine-grained spaces are indeed supersets of the coarse ones, their size grows exponentially, making it substantially more challenging to search effectively within fixed parameter budgets. This explains why coarser spaces can perform better under a reasonable search budget.
>
> More sample-efficient methods e.g., Bayesian optimization [1] or incorporating architectural priors within bins [2] could improve search in larger spaces. However, as our primary goal with whittle is to provide a modular library for SLM search and pretraining, these extensions are beyond the current scope of our work.
>
> **Search Space Sizes (N) for Pythia-6.9B and Pythia-12B Architectures**
>
> | Search Space               | Pythia-6.9B (N)          | Pythia-12B (N)          |
> |---------------------------|---------------------------|--------------------------|
> | **Coarse Uniform (CU)**   | 4.33 × 10¹²               | 9.51 × 10¹²              |
> | **Coarse Layer-wise (CL)**| 1.65 × 10²⁴⁴              | 2.33 × 10²⁸¹             |
> | **Fine-grained Uniform (FU)** | 10¹⁴²⁸⁴               | 10¹⁷⁸⁴¹                 |
> | **Fine-grained Layer-wise (FL)** | 1.58 × 10¹⁵⁹⁹⁸⁴   | 5.31 × 10²²⁴⁶¹¹          |
>
> We would like to thank you again for the detailed review. We hope that we were able to address your concerns and that you would consider increasing your score. We are happy to engage in further discussions with you and address any further questions you may have during the rebuttal period.
>
> [1] White, C., Neiswanger, W. and Savani, Y., 2021, May. Bananas: Bayesian optimization with neural architectures for neural architecture search. In Proceedings of the AAAI conference on artificial intelligence (Vol. 35, No. 12, pp. 10293-10301).
>
> [2] Mallik, N., Bergman, E., Hvarfner, C., Stoll, D., Janowski, M., Lindauer, M., Nardi, L. and Hutter, F., 2023. Priorband: Practical hyperparameter optimization in the age of deep learning. Advances in Neural Information Processing Systems, 36, pp.7377-7391.

---

> > ### Author Response · Authors · 2025-11-26
> > **Comparing random search with evolutionary search**
> >
> > As requested by the reviewer, we have now performed random search as a baseline for each of the bins. The best model obtained from this search phase was then trained with 2 billion tokens. The results can be found [here](https://figshare.com/s/334e07f34c6b0aa65d9a). The models obtained with evolutionary search beat those found by random search on two out of the three bins. Furthermore, we would like to emphasize that the evolutionary search procedure itself is not the primary contribution of our work. Evolutionary search is merely one instantiation of a generic search algorithm, which we found to perform well in our experiments, and can be replaced by alternatives such as random search or Bayesian optimization. Our central contribution is **whittle**, a flexible library that enables users to extract effective SLMs from pretrained LLMs. The framework is designed so that users can easily plug in more sophisticated search strategies, which may improve results further.
> >
> > Thank you again for your suggestion and we are happy to engage in further discussions with you until the end of the rebuttal period.

---

### Author Response · Authors · 2025-11-26
**Additional Results: Whittle SLMs trained for 100B token budget**

To understand how the cost savings from subnetwork extraction scale with larger pretraining budgets, we trained the best model in bin 3 for 100 billion tokens and compared it to Pythia-2.8B trained for the same number of tokens from random initialization. We make three observations:
Our model reaches the same validation performance while requiring **1.26x fewer FLOPs (or ~21% fewer FLOPS)**. Although the reduction is smaller than the **5.16x** savings as observed at the 10-billion-token scale, it shows that subnetwork extraction continues to provide efficiency benefits even when the training budget is increased by an order of magnitude. This indicates that the method scales beyond small-budget regimes and remains competitive at larger compute settings.

Our model achieves a lower final validation perplexity. While the Pythia model has a validation perplexity of 11.397, our model has 11.204. Furthermore, our model outperforms the base Pythia models trained from random initialization by a significant margin by up to 12% on MMLU-cont (check table below).

| Base Model   | Initialization | #Params | COPA | OpenBookQA | Lambada-OpenAI | Winogrande | Social IQA | MMLU-cont. | MMLU  | CommonsenseQA | PIQA  | ARC-challenge | ARC-easy | HellaSwag | BoolQ | Avg-acc |
|--------------|----------------|---------|------|------------|----------------|------------|------------|-------------|-------|----------------|-------|----------------|----------|-----------|--------|---------|
| Pythia-6.9B  | Supernet Init  | 1.04B   | **74.00** | **38.40** | **45.604**      | **56.98**   | 41.81     | **32.54**    | **27.04** | **20.88**       | **75.03** | **38.57**       | **71.76**  | **60.05**  | **61.19** | **49.53** |
| Pythia-1B    | Random Init    | 1.01B   | 71.00 | 35.40     | 36.13         | 53.35      | **41.91**   | 26.93      | 25.20 | 19.74         | 73.50 | 35.92         | 69.36   | 55.69    | 52.75 | 45.91 |
| Pythia-6.9B  | Supernet Init  | 2.91B   | **76.00** | 37.40     | **53.23**       | 58.01      | **42.02**   | **34.45**    | **38.04** | **43.41**       | **77.69** | **41.38**       | **73.57**  | **64.01**  | **63.49** | **54.05** |
| Pythia-2.8B  | Random Init    | 2.78B   | 71.00 | **40.40**  | 43.39         | **58.64**   | **42.02**   | 27.18      | 25.74 | 21.37         | 76.50 | 42.75         | 74.24   | 64.85    | 60.58 | 49.23 |

We will include these results in the paper and update the abstract accordingly.

---

### Author Response · Authors · 2025-11-27
**Updates to paper and summary of changes**

As the discussion period closes soon, we would once again like to thank all the reviewers for thoroughly reviewing our work and engaging with us during the discussion period. Their feedback on our work helped us greatly improve the writing of the paper and our experimental evaluation. We have now updated our paper, with all major changes highlighted in red. We provide a summary of main changes to the paper below:

1. **Abstract and Introduction**: We have updated the introduction and abstract of the paper to reflect the saving factors after inclusion of the evolutionary search cost (as recommended by reviewers 4Dsu, HjZw, KKnz and 5QXR) .
2. **Methodology**: We have improved the description Section 2.2 with more details on the crossover and mutation procedures (as suggested by reviewer HjZw)
3. **Evolutionary Search Setup**: We have given further details on setup for the evolutionary search algorithm in Section 4.2 (as requested by reviewer HjZw)
4. **Appendix A** : To the related work we have now added discussion on comparison to Sheared Llama and DRPruning (as suggested by reviewer 5QXR).
5. **Appendix B.2**: We have added details on the computational cost overhead introduced by evolutionary search (as suggested by reviewers 4Dsu, KKny, and 5QXR)
6. **Appendix C.1**: We have provided more details on the searching for heads and head size for different transformer attention types, showing generality of our library whittle.
7. **Appendix E**: We provide details on the importance scoring procedure from Minitron [1] implemented in whittle, which we use for search in Section 5 ablation.
8. **Appendix G**: We provide an analysis on the importance of different hyperparameters when performing logit based distillation, offering interesting insights into currently choosing distillation hyperparameters.
9. **Appendix H**: We report the search space sizes to show the exponential search space size for the finegrained spaces (as per discussion with reviewer 4Dsu).
10. **Appendix J**: We provide additional results when scaling pretraining to a token budget of 100B tokens.

If the reviewers have any further questions, we are happy to discuss with them until the discussion period closes.

---

### Author Response · Authors · 2025-12-04
**Summary of Reviews, Revisions, and Additional Analyses**

Dear Reviewers, ACs, and SACs,

With the discussion period closing soon, we would like to sincerely thank all reviewers for their positive and encouraging feedback: **Reviewer 4Dsu** (**score 6**)  highlighted that our paper is well written, with a clearly defined methodology and goal, and found the evolutionary search intuitively presented; **Reviewer KKny** (**score 8**)  recognized the novelty and convincing nature of our idea, praised the thoroughness of our evaluation and the impressive speedups, and especially valued our released open-source, reproducible framework *whittle* as a primary contribution; **Reviewer 5QXR** (**score 4**) appreciated the strengths of our approach in enabling the training of SLMs under limited compute budgets and the practical usefulness of the *whittle* library for extracting SLMs from open-source LLMs; and **Reviewer HjZw** (**score 4**) emphasized several positive aspects, including our insights on distillation for SLMs and the systematic exploration of search spaces at different levels of granularity.

The suggestions and feedback from the reviewers greatly helped us improve the quality of our paper and our evaluation setup. Notably, reviewers **4Dsu**, **KKny**, and **5QXR** pointed out that the estimates for compute savings compared to the baseline should also account for the cost of the evolutionary search. We agree and have updated the numbers, taking the search cost into consideration. We have also since then updated the manuscript to elaborate on the evolutionary search mechanism in more depth, covering details such as search cost and the methods of mutation and crossover in the space of the language models, and improved the section covering related work. Further, we trained a model with a 100B token budget to test whether the benefits of extracting a subnetwork from the pretrained supernet persist at larger scales. We found that the 2.8B model extracted from Pythia-6.9B requires **21% fewer FLOP** to match the validation perplexity of a Pythia-2.8B model pretrained on the same number of tokens. This model was trained with 10x more tokens than the largest experiments that were originally present in the manuscript.

We would also like to update the reviewers on an additional analysis conducted during the rebuttal period. Motivated by the reviewer feedback and our experiments, we fit a power law to the compute savings observed across different token budgets.  We find that the compute-savings factor generally increases as the token budget grows across the range we evaluate (10B–100B tokens). Although there is some variability in the measured factors, the overall trend and the fitted power-law model both indicate that substantial compute savings persist for SLMs of this size across diverse token budgets. We provide the corresponding figure with the power-law fit [here](https://figshare.com/s/dba1ed0400ab8b427056?file=60086792).

Lastly, we compare the perplexity of pruned architectures extracted from Llama-3.1-8B using our evolutionary search (without any pretraining or fine-tuning) against architectures obtained via structured pruning methods such as LLMPruner [1] (also without any pretraining or fine-tuning). We observe that our expressive search space combined with a simple evolutionary search consistently yields architectures with **substantially lower perplexity on WikiText at initialization, while also using fewer parameters**. We report the WikiText perplexity of the extracted LLaMA subnetworks at initialization in the table below. These results further demonstrate that **whittle**, as a framework for extracting high-quality subnetwork initializations, generalizes effectively to other LLM families beyond Pythia.

### Comparison by Parameter Regime (SLMs without additional pretraining)

| Param Regime (~B) | Method     | Param Count (B) | Perplexity on WikiText     |
|-------------------|------------|------------------|----------------|
| ~1.5–2.0B         | Ours       | **1.447**        | **11789.92** |
| ~1.5–2.0B         | LLMPruner  | 1.908            | 13359.73   |
| ~2.5–2.8B         | Ours       | **2.585**        | **8625.69**  |
| ~2.5–2.8B         | LLMPruner  | 2.725           | 11882.39     |


[1] Ma, X., Fang, G. and Wang, X., 2023. Llm-pruner: On the structural pruning of large language models. Advances in neural information processing systems, 36, pp.21702-21720.

We hope our responses adequately address the reviewers’ concerns and help clarify the contributions and significance of our work.

---

### Meta-Review · Area_Chair_BxfC · 2026-01-05

**Summary:**

The goal of this paper is to compress an LLM into an SLM. Its contributions are, as listed in the abstract and intro:
- Smart initialization for the SLM weights by taking sub-networks from the LLM
- An evolutionary-search method (ES) to navigate the space of initializations, where the objective is to have as low perplexity as possible on a training set.
- Distillation of the LLM into the SLM using knowledge distillation
- A library, Whittle, that allows us to do all of this seamlessly.

The reviewers praised the writing of the work, the simplicity of the method, and the release of the code.
The main concerns of the reviewers are:
- **Novelty** (HjZw, 5QXR): this is not the first work trying to distill an LLM into an SLM, and the baselines implemented in this work are not representative of the state of the art.
- **Search cost** (4dsu, HjZw, KKny, 5QXR): the cost of search should be taken into account and compared to the cost of pretraining when comparing methods.

While the authors sought to address these concerns in the rebuttal, I believe they remain outstanding, and the paper's contributions are insufficient at this stage. In order to improve the paper, the authors could make several comparisons to baselines more convincing, and center the paper around the features of whittle.



Misc remarks:
- Fig 1 is too small, fonts should be same size as main text.
- I think that a discussion about inference speed would strenghten the paper. The paper focuses on model size, and while it is a good proxy of inference speed, models with same total number of parameters but different architecture may have different inference speeds.
- Model name changes halfway through, it goes from T to M in 2.2
- fig 3 overlaps with text
- in fig 3, it would be great to report the perplexity of random initializations to see how ES helps.
- What happens when distilling a random init model in fig 5?
- fig 7 seems orthogonal to the rest of the paper. The fact that using only top k logits for distillation leads to a trade-off between compute and perplexity is well known.

**Reviewer Concerns:**

# Novelty / other baselines

The authors conduct the following ablations to compare to baselines, post-rebuttal:
- ES (proposed) vs random search: ES is not consistently better than random search
- Perplexity (proposed) vs other criteria (activation-based importance scores such as for Minitron and weight-magnitude scores) as targets for search (new fig 8.) The authors show that the models obtained have a lower perplexity at initialization. However, these models are not further trained; it would be beneficial to know if the gains of initialization persist over training. This crucial information is missing from the paper.

The fact that distillation then improves the proposed models is not novel, several previous works already demonstrate that.
Overall, only the initialization strategies seem to be novel and bring an improvement compared to baselines.

The authors brush aside a direct comparison to shearable Llama because it requires backprop, but the proposed method computes perplexity, which also requires a lot of compute (a non-zero fraction of the training budget as seen in the rebuttal). Also, the model is then trained anyways, which requires backprop. It would therefore be good to properly evaluate the method against shearable LLAMA, taking into account the initialization cost.

In the rebuttal, the authors argue that the fact that these gains are not meaningful is counterbalanced by the fact that their "central contribution is whittle, a flexible library that enables users to extract effective SLMs from pretrained LLMs." All reviewers agree that this is an important contribution, but *the paper is currently not written in such a way*. Only one page is dedicated to the library per-se, and the paper is not written as a way to demonstrate whittle's features. The abstract and introduction clearly mention new methodological advances, but the current paper does not convincingly demontrate them.


# Flops-corrected curves

The authors have computed the cost of the search conducted in their experiments; for all model sizes, it seems to be roughly the same cost as training for 1B tokens. The authors have then updated their claims about efficiency, showing that be get a 5x improvement compared to random initializations when training for 10B tokens.
However, the acceleration factor depends strongly on the number of tokens used to train. For instance, in fig.4, left, it seems like training a random model for 1B tokens leads to a perplexity of ~100 (probably lower if the cosine decay ends at that point), which is far lower than the perplexity obtained after the supernet init (~5k). Hence, under a low compute budget, one is better off pretraining a random model than using the proposed search method. This critical fact should be acknowledged.

# Use of other base models

The authors present the results of the proposed method using llama 3.1, against llm-pruner. This improves the paper a lot. However,
the authors only show improvements to the initialization perplexity, and like in fig 8, do not demonstrate that these gains persist over training. This important experiment is missing.

**Reviewer Scores:**

- 4dsu: I think that the authors only gave a partial answer to the flops-normalized question, since the provided acceleration factors strongly depend on the training budget. The added comparison to random search makes it clear that the proposed ES method is not much better. I think that reviewer would have stayed at 6.
- HjZw: The main answer to this reviewer's criticism is that the paper is about the library. However, this is not really the case at that stage; there are claims of methodological advances in the paper. I think this reviewer would have kept it a 4.
- KKny: This reviewer was very happy with the current paper, and the authors responded clearly to their concerns. I think they would have stayed an 8.
- 5QXR: same as HjZw, I think they would have kept it a 4.

---

### Decision · Program_Chairs · 2026-01-26

Reject